

# The STRatospheric Estimation Algorithm from Mainz (STREAM): Estimating stratospheric NO$_2$ from nadir-viewing satellites by weighted convolution

S. Beirle [1], C. Hörmann [1], P. Jöckel [2], M. Penning de Vries [1], A. Pozzer [1], H. Sihler [1], P. Valks [3], and T. Wagner [1]

[1]Max-Planck-Institut für Chemie (MPI-C), Mainz, Germany
[2]Deutsches Zentrum für Luft- und Raumfahrt (DLR), Institut für Physik der Atmosphäre, Oberpfaffenhofen, Germany
[3]Deutsches Zentrum für Luft- und Raumfahrt (DLR), Institut für Methodik der Fernerkundung (IMF), Oberpfaffenhofen, Germany

*Correspondence to:* S. Beirle
steffen.beirle@mpic.de

**Abstract.** The STRatospheric Estimation Algorithm from Mainz (STREAM) determines stratospheric columns of NO$_2$ which are needed for the retrieval of tropospheric columns from satellite observations. It is based on the total column measurements over clean, remote regions as well as over clouded scenes where the tropospheric column is effectively shielded. The contribution of individual satellite measurements to the stratospheric estimate is controlled by various weighting factors. STREAM is
a flexible and robust algorithm and does not require input from chemical transport models. It was developed as verification algorithm for the upcoming satellite instrument TROPOMI, as complement to the operational stratospheric correction based on data assimilation. STREAM was successfully applied to the UV/vis satellite instruments GOME 1/2, SCIAMACHY, and OMI. It overcomes some of the artefacts of previous algorithms, as it is capable of reproducing gradients of stratospheric NO$_2$, e.g. related to the polar vortex, and reduces interpolation errors over continents. Based on synthetic input data, the un-
certainty of STREAM was quantified as about 0.1-0.2 $\times 10^{15}$ molecules cm$^{-2}$, in accordance to the typical deviations between stratospheric estimates from different algorithms compared in this study.

## 1   Introduction

Beginning with the launch of the Global Ozone Monitoring Experiment (GOME) on the ERS-2 satellite in 1995 (Burrows et al., 1999), several instruments perform spectrally resolved measurements of sunlight reflected by the Earth's surface and atmo-
sphere (Table 1). With differential absorption spectroscopy (DOAS), the column densities (denoted as "columns" henceforth) of numerous important atmospheric absorbers can be determined by their characteristic spectral "fingerprints" (Platt and Stutz, 2008), amongst others nitrogen dioxide (NO$_2$).

   Nitrogen oxides (NO$_x$=NO$_2$+NO) play a key role in the chemistry of both the stratosphere and the troposphere. Stratospheric NO$_x$ has been a research topic for several decades, in particular due to its role in ozone and halogen chemistry.



Satellite measurements provide long-term global information on spatio-temporal patterns of stratospheric $NO_2$ (e.g., Wenig et al., 2004; Dirksen et al., 2011). During the last decades, the analysis of tropospheric trace gases from nadir-viewing satellite instruments moved more and more into focus, supported by the availability of longer time-series and improved spatial resolution. Tropospheric $NO_2$ columns derived from satellite are nowadays widely used by the scientific community to deduce spatial patterns, source type and strength, and trends of $NO_x$ emissions from fossil fuel combustion, biomass burning, soil emissions, and lightning. Overviews over the wide range of scientific applications of satellite-based tropospheric $NO_2$ products are given in e.g. Martin (2008) or Monks and Beirle (2011).

The retrieval of tropospheric $NO_2$ columns from total column measurements requires the estimation and removal of the stratospheric column, a procedure we refer to as "Stratosphere-Troposphere-Separation" (STS) as in Bucsela et al. (2006).

One of the first STS algorithms is the reference sector method (RSM), which estimates the global stratospheric $NO_2$ fields from measurements over the remote Pacific (Richter and Burrows, 2002; Martin et al., 2002; Beirle et al., 2003), based on the assumptions of (a) longitudinal homogeneity of stratospheric $NO_2$, and (b) negligible tropospheric contribution over the reference region in the Pacific. This procedure is quite simple, transparent, and robust. A further side effect is that any systematic bias in the $NO_2$ columns, which might be introduced by the instrument (like e.g. degradation, or spectral interference caused by the diffusor plate used for measurements of the solar reference (Richter and Burrows, 2002)) or sub-optimal spectral analysis (van Geffen et al., 2015; Marchenko et al., 2015), is classified as stratospheric signal, and thereby removed from the tropospheric column.

The RSM was applied by different groups to different satellite instruments and generally performs well. However, the resulting tropospheric $NO_2$ columns are affected by systematic biases caused by the simplifying assumptions:

(a) The tropospheric background column in the Pacific is very low (compared to columns over regions exposed to significant $NO_x$ sources), but not 0. Neglecting the tropospheric background results in tropospheric columns that are biased low by about $0.1 \times 10^{15}$ molec cm$^{-2}$ (Valks et al., 2011). Some algorithms explicitly correct for this tropospheric background: Martin et al. (2002) perform a correction based on GEOS-CHEM, while Valks et al. (2011) assume a constant background of $0.1 \times 10^{15}$ molec cm$^{-2}$. Other algorithms prefer to stick to the tropospheric "excess" columns, which are slightly biased low, but do not need any model input (Richter and Burrows, 2002; Bucsela et al., 2006).

(b) The assumption of longitudinal homogeneity is generally reasonable, at least in temporal means when small scale stratospheric dynamic features cancel out. But in particular close to the polar vortex, large longitudinal variations can occur, as already discussed by Richter and Burrows (2002), Martin et al. (2002) or Boersma et al. (2004). Thus, tropospheric columns derived by RSM can be off by more than $10^{15}$ molec cm$^{-2}$ in winter at latitudes from $50°$ polewards, thereby affecting scientific interpretations of tropospheric columns over North America or Northern Europe. Note that also at low latitudes, systematic artefacts show up in tropospheric columns resulting from RSM, in particular over the Indian ocean, which are related to longitudinal inhomogeneities.

To overcome the artefacts caused by the assumption of longitudinal homogeneity, several modifications of the RSM have been proposed in recent years, while the basic approach of using nadir measurements over clean regions for STS has been retained. We refer to this group of algorithms as "modified RSM" (MRSM). MRSMs generally define a "pollution mask"





of regions with potentially non-negligible tropospheric columns. Measurements over these regions are skipped within the stratospheric estimation. Thus, in order to define stratospheric columns over the masked areas, interpolation is required. For this purpose, Leue et al. (2001) and Wenig et al. (2004) applied "normalized convolution" (Knutsson and Westin, 1993), an efficient algorithm which combines interpolation and smoothing. Bucsela et al. (2006) realized interpolation by fitting

harmonics (wave-2) over the "clean" areas. Valks et al. (2011) applied a zonal boxcar filter of 30°.

All of these algorithms applied a rather conservative masking approach for potentially polluted pixels. Continents were masked out almost completely. At Northern mid-latitudes, the masked area is often even larger than the area used for the stratospheric estimation, and over the Eurasian continent, the MRSM algorithms miss any supporting measurement points over about ten thousand km. This can lead to significant errors during interpolation. In particular the wave fitting approach can lead

to large biases (Dirksen et al., 2011).

Leue et al. (2001) estimated the stratospheric fields based on clouded measurements over the ocean and subsequent interpolation. The focus on clouded observations provides a direct stratospheric measurement, as the tropospheric column is mostly shielded; thus, no further correction of the tropospheric background should be needed. On the other hand, clouded pixels possibly contain $NO_x$ produced by lightning (e.g., Beirle et al., 2006). Therefore, Wenig et al. (2004) changed the Heidelberg STS

algorithm (Leue et al., 2001) by switching from clouded to cloud free observations as input for the stratospheric estimate[1].

Recently, Bucsela et al. (2013) proposed a MRSM which defines "unpolluted" pixels not with a fixed mask, but according to the a-priori expected tropospheric contribution to the total column for each individual satellite observation. This is determined from radiative transfer calculations based on a monthly mean $NO_2$ profile from a chemical transport model (CTM) and the actual cloud conditions. This procedure results in additional supporting points over continents in cases of clouds shielding the

tropospheric column and thereby largely reduces potential interpolation artefacts.

Apart from (modified) reference sector methods, there are two further completely different approaches used for STS which are based on (a) independent measurements, or (b) CTMs.

(a) Coincident, but independent stratospheric measurements are available for the SCanning Imaging Absorption Spectro-Meter for Atmospheric CHartographY SCIAMACHY (Bovensmann et al., 1999). It was operated in alternating nadir/limb

geometry, such that the stratospheric air masses sensed in nadir were scanned in limb shortly before ("limb-nadir-matching", LNM) . This unique instrumental set-up allowed for a direct stratospheric correction, although systematic offsets between limb and nadir measurements still had to be corrected empirically. STS by LNM was successfully applied for $NO_2$ (Sioris et al., 2004; Sierk et al., 2006; Beirle et al., 2010a; Hilboll et al., 2013) and ozone (Ebojie et al., 2014). However, such direct coincident measurements of total columns (nadir) and stratospheric concentration profiles (limb) are not available for

other satellite instruments, and merging measurements from different sensors always faces the problem of spatio-temporal mismatching, requiring interpolation and photochemical corrections (compare Belmonte Rivas et al., 2014), and thus cannot be easily used for consistent long-term operational retrievals.

---

[1]This aspect will be discussed in detail in section 5.4.





(b) Stratospheric $NO_2$ concentrations provided by CTMs can be used directly for STS after empirical correction of systematic offsets between satellite and model columns, e.g. by matching both over the Pacific (Richter et al., 2005; Hilboll et al., 2013).

A more sophisticated way to incorporate CTMs in STS is data assimilation (Eskes et al., 2003; Dirksen et al., 2011), in which modelled 3D distributions of $NO_2$ are regularly updated such that the modelled stratospheric column is in close agreement with the satellite measurement if the tropospheric contribution (as forecasted by the CTM) is low.

In 2016, the ESA's Sentinel 5 precursor (S5p) satellite (Ingmann et al., 2012) is going to be launched, carrying the TROPOspheric Monitoring Instrument TROPOMI (Veefkind et al., 2012). The operational ("prototype") tropospheric column product of $NO_2$ from TROPOMI will be derived by a STS using data assimilation (van Geffen et al., 2014), based on the expertise of the Koninklijk Nederlands Meteorologisch Instituut (KNMI) as demonstrated by a 20 year record of tropospheric columns from different satellite sensors provided by the Tropospheric Emission Monitoring Internet Service (TEMIS, www.temis.nl) (Boersma and Eskes, 2004; Boersma et al., 2011)).

Within the S5p Level2 project, for each prototype product a "verification" product was developed in order to verify the prototype algorithms, detect possible shortcomings and reveal potential improvements. The TROPOMI verification algorithm for $NO_2$ STS, the "STRatospheric Estimation Algorithm from Mainz" STREAM, was developed at the Max-Planck-Institut für Chemie (MPI-C), Mainz. It is a MRSM, requiring no further model input, and can thus be considered as a complimentary approach to data assimilation.

STREAM does not apply a strict discrimination of "clean" versus "polluted" satellite pixels. Instead, weighting factors are defined for each satellite pixel determining its impact on the stratospheric estimate (similar as in data assimilation). In particular, clouded observations are weighted high, as they provide direct measurements of the stratospheric field.

The manuscript is organized in the following way:

In section 2, the STREAM algorithm is described in detail. Section 3 provides information on the satellite and model datasets used in this study. Section 4 analyses the performance of STREAM and its sensitivity to input parameters based on both, actual satellite measurements and synthetic data. In section 5, the STREAM results are discussed in comparison to other STS algorithms, including the TROPOMI prototype algorithm. A general discussion on the challenges and uncertainties of STREAM in particular, and STS in general, is given, followed by conclusions (section 6). Several additional images and tables are provided in a supplementary document.

## 2 Methods

STREAM stands in tradition of MRSM algorithms which estimate the stratospheric field directly from satellite measurements for which the tropospheric contribution is considered to be negligible. For this purpose, measurements over remote regions with negligible tropospheric sources, as well as cloudy measurements are used. In contrast to other MRSMs, however, no strict pollution mask is applied. Instead, weighting factors are used.





STREAM consists basically of two steps:

1. a set of weighting factors are calculated for each satellite pixel, determining to what extent the measured $NO_2$ total columns contribute to the estimated stratospheric field (Sect. 2.2), and

2. global maps of stratospheric $NO_2$ are determined by applying weighted convolution (Sect. 2.3).

Before describing the details of the STREAM algorithm, however, we first define the investigated quantities and abbreviations used hereafter, as summarized in Table 2.

## 2.1    Terminology

### 2.1.1    Supplementary Material

Additional images, tables and text are provided in a supplementary document. All references to tables and figures in the
supplement are indicated by a prefix "S". For readability, the supplement is structured analogously to the manuscript. I.e., additional material to section 2.3 can be found in section S2.3 of the supplement.

### 2.1.2    $NO_2$ column densities and units

With Differential Optical Absorption Spectroscopy (DOAS, Platt and Stutz (2008)), so-called slant column densities (SCDs) $S$, i.e. concentrations integrated along the mean light path, are derived. SCDs are converted into VCDs (vertical column densities,
i.e. vertically integrated concentrations) $V$ via the air-mass factor (AMF) $A$: $V = S/A$. The AMF $A$ depends on radiative transfer (determined by wavelength, atmospheric absorbers, viewing geometry, surface albedo, clouds, and aerosols) and the trace gas profile. For the stratospheric column of $NO_2$, $A$ is basically determined by viewing geometry.

In this study, all column densities are given in column density units (CDU) to increase readability:

$$1 \text{ CDU} := 1 \times 10^{15} \text{ molecules cm}^{-2}. \tag{1}$$

### 2.1.3    Total vertical column $V^*$

We define $V^*$ as "total" vertical column, given by the SCD $S$ divided by the stratospheric AMF $A_{\text{strat}}$:

$$V^* = S/A_{\text{strat}}, \tag{2}$$

The application of the stratospheric AMF basically removes the dependencies of $S$ on viewing angles. Over clean regions with negligible tropospheric columns, $V^*$ represents the actual total VCD, and can be used for the estimation of stratospheric fields.
In case of tropospheric pollution, however, $V^*$ underestimates the actual total VCD, as the AMF is generally smaller in the troposphere than in the stratosphere (see also next section). These situations are, to the best of knowledge, excluded from the stratospheric estimate by the definition of appropriate weighting factors (see sect. 2.2).





### 2.1.4 Stratospheric vertical column and tropospheric residue

STREAM yields an estimate for the stratospheric VCD $V_{\text{strat}}$ based on the assumption that $V^*$ can be considered as proxy for $V_{\text{strat}}$ in "clean" regions and over cloudy scenes.

In order to evaluate the performance of the stratospheric estimation, we define the tropospheric residue (TR) as the difference

of total and stratospheric VCDs (based on a stratospheric AMF):

$$T^* = V^* - V_{\text{strat}}, \tag{3}$$

Tropospheric VCDs (TVCDs), which are the final product of $NO_2$ retrievals used for further tropospheric research, are connected to $T^*$ via the ratio of stratospheric and tropospheric AMF:

$$V_{\text{trop}} = T^* \times \frac{A_{\text{strat}}}{A_{\text{trop}}}. \tag{4}$$

For cloud-free satellite pixels, the ratio $A_{\text{strat}}/A_{\text{trop}}$ typically ranges from about 1 above clean oceans at low and mid-latitudes to $\approx$ 2-3 above moderately polluted regions, and up to >4 at high latitudes and over strong $NO_x$ sources, where $NO_2$ profiles peak close to the ground, causing low $A_{\text{trop}}$. Figure S1 in the supplement displays monthly mean ratios $A_{\text{strat}}/A_{\text{trop}}$ for cloud free scenes based on AMFs provided in the NASA OMNO2 product.

In this study, we focus on the tropospheric residue $T^*$ instead of $V_{\text{trop}}$ for several reasons:

1. As only stratospheric AMFs are applied, biases in the stratospheric estimation can directly be related (factor -1) to the respective biases in $T^*$.

2. The comparison of TRs among different algorithms instead of TVCDs isolates the effect of the different STS and excludes differences in tropospheric AMFs (which are beyond the scope of this study), and

3. $T^*$ can be determined and is of high interest for the evaluation of STS performance also for clouded scenes with very low

tropospheric AMFs.

### 2.1.5 Version

The description given in this manuscript and the definition of a-priori settings refer to STREAM version v0.92.

## 2.2 Definition of weighting factors

MRSMs usually flag satellite pixels as either clean or (potentially) polluted, and skip the latter for the stratospheric estima-

tion. In STREAM, instead, weighting factors for individual satellite pixels determine how strongly they are considered in the stratospheric estimation. Satellite measurements which are expected to have low/high tropospheric contribution are assigned a high/low weighting factor, respectively.

### 2.2.1 Pollution weight

In order to estimate the stratospheric $NO_2$ field from total column measurements, only "clean" measurements where the

tropospheric column is negligible, should be considered. In cases of very high total columns ($V^*$>10 CDU) which clearly





exceed the domain of stratospheric columns, a tropospheric contribution is obvious, and these measurements are excluded by assigning them a weighting factor of 0.

In most cases, however, the tropospheric contribution to the total column is not that easy to determine. We thus define a pollution weight $w_{\mathrm{pol}}$ based on our a-priori knowledge about the mean spatial distribution of tropospheric $NO_2$, reflecting the tropospheric pollution probability. For this purpose, we make use of the multi-annual mean tropospheric $NO_2$ column as derived from SCIAMACHY (Beirle and Wagner, 2012). Based on this climatology, a "pollution proxy" $P$ is defined as function of latitude $\vartheta$ and longitude $\varphi$. $P$ indicates the regions affected by tropospheric pollution plus a "safety margin" in order to account for possible advection, while it is undefined for remote unpolluted regions. Details on the definition of $P$ are given in the supplement (sect. S2.2.1).

The pollution weight $w_{\mathrm{pol}}$ is then defined as

$$w_{\mathrm{pol}} = 0.1/P(\vartheta,\varphi)^3, \tag{5}$$

where $P$ is defined, and $w_{\mathrm{pol}} = 1$ elsewhere. Hence, the higher the pollution proxy $P$, the lower the weighting factor and the less the measurement contributes to the stratospheric estimate. Equation 5 is displayed in Fig. 1(a), and the resulting map for $w_{\mathrm{pol}}$ is shown in Fig. 2(a). Large continental regions are assigned with a weight $<=0.1$. Strongly polluted regions like the US, Europe, or China have weights of 0.01 down to below 0.001. Note that the additional application of the tropospheric residue weight (Sect. 2.2.3) further decreases the weight of satellite measurements containing high tropospheric pollution.

### 2.2.2 Cloud weight

In addition to measurements over remote regions free of tropospheric sources, also clouded satellite measurements, where the tropospheric column is shielded, provide a good proxy for the stratospheric column. Thus, the factor $w_{\mathrm{cld}}$ is used to increase the weight of clouded satellite pixels. This is achieved by the following definition:

$$
\begin{aligned}
w_{\mathrm{cld}} \quad &:= 10^{2 \times w_c \times w_p} && \text{(a)} \\
\text{with} \quad w_c \quad &:= c^4 && \text{(b)} \\
\text{and} \quad w_p \quad &:= e^{-\frac{1}{2}\left(\frac{p_{\mathrm{cld}} - p_{\mathrm{ref}}}{\varsigma_p}\right)^4} && \text{(c)}
\end{aligned}
\tag{6}
$$

$w_c$ reflects the dependency on the cloud radiance fraction (CRF) $c$. Due to the exponent of 4, only pixels with large cloud radiance fraction obtain a high weighting factor and contribute strongly to the stratospheric estimation.

$w_p$ describes the dependency on cloud pressure (CP) $p_{\mathrm{cld}}$. It is defined as a modified Gaussian (with exponent 4 instead of 2, making it flat-topped) centered at $p_{\mathrm{ref}} = 500$ hPa with the width $\varsigma = 150$ hPa. I.e., only cloudy measurements at medium altitudes are assigned a high weighting factor, while high clouds (potentially contaminated by lightning $NO_x$) as well as low clouds (where tropospheric pollution might still be visible) are excluded.

As both $w_c$ and $w_p$ yield values in the range from 0 to 1, the factor of 2 in the exponent of Eq. 6(a) sets the maximum value of $w_{\mathrm{cld}}$ to $10^2$, which would compensate for pollution weights down to $10^{-2}$.




The dependencies of $w_{\mathrm{cld}}$ on CRF and CP, as defined in Eq. 6, are displayed in Fig. 1 (b) and (c), respectively. The spatial pattern of $w_{\mathrm{pol}}$ is shown exemplarily for OMI CP and CRF on 1 January 2005 in Fig. 2(b). $w_{\mathrm{cld}}$ reaches values up to 100 in several parts of the world, including regions which were pre-classified as potentially polluted, thus competing with a low $w_{\mathrm{pol}}$ (Fig. 2(a)).

**2.2.3   Tropospheric residue weight**

STREAM yields global fields of stratospheric VCDs $V_{\mathrm{strat}}$, explained in detail below (sect. 2.3), which allow to calculate tropospheric residues $T^*$ according to Eq. 3. While the "true" tropospheric fields are not known, the resulting $T^*$ can still be used in order to evaluate the STS performance and improve the stratospheric estimate in a second iteration, whenever $T^*$ clearly indicates an under- or overestimation of $V_{\mathrm{strat}}$:

– A high value of $T^*$ generally indicates tropospheric pollution. The respective satellite pixels should not be used for the stratospheric estimation.

  – As negative columns are nonphysical, $T^*<0$ indicates that the stratospheric field has been overestimated. Consequently, the respective satellite measurements should be assigned a higher weighting factor such that they contribute more strongly to the stratospheric estimate.

We thus define a further weighting factor $w_{\mathrm{TR}}$ which weights down/up the pixels associated with a large positive/negative TR, respectively. It turned out, however, that the stratospheric estimate is very sensitive to the definition of $w_{\mathrm{TR}}$, and a simple definition based on the TR of individual satellite pixels can easily result in systematic artefacts. This results from $T^*$ being defined as the difference of $V^*$ and $V_{\mathrm{strat}}$ (Eq. 3), i.e. two quantities of the same order of magnitude with non-negligible errors. Thus, the resulting statistical distribution of $T^*$ inevitably includes negative values. These negative values caused by statistical

fluctuations must not be excluded from the probability density function in order to keep the mean unbiased, but should not be used as trigger for weighting up the respective measurement within the stratospheric estimation. Thus, $w_{\mathrm{TR}}$ should be only applied to significant and systematic deviations of $T^*$ from 0. This is achieved by the following settings:

  1. In contrast to $w_{\mathrm{cld}}$, which is defined for each individual satellite measurement, $w_{\mathrm{TR}}$ is defined based on the TRs averaged over $1° \times 1°$ grid pixels. I.e., first the values of $T^*$ within one grid pixel are averaged, reducing statistical noise, before

$w_{\mathrm{TR}}$ is calculated, and the resulting weight is then assigned to all satellite measurements within the grid pixel.

  2. $w_{\mathrm{TR}}$ is only applied if the absolute value of the mean grid box $T^*$ exceeds a threshold of 0.5 CDU, which is typically larger than the spectral fitting error:

$$w_{\mathrm{TR}} := \begin{cases} 10^{-2 \times T^*} & \text{if } |T^*| > 0.5 \text{ CDU} \\ 1, & \text{else} \end{cases} \tag{7}$$

  3. $w_{\mathrm{TR}}$ is only applied, if a larger area is affected by systematic low or high TR, i.e. if the adjacent grid pixels exceed the

threshold as well. I.e., a single outlier will not trigger $w_{\mathrm{TR}}$.





$w_{\mathrm{TR}}$ could in principle be tuned in multiple iterations. In STREAM v0.92, only one iteration is performed, as a second iteration turned out to have marginal effect (see section S4.2.3).

The dependency of $w_{\mathrm{TR}}$ on TR (grid pixel average), as defined in Eq. 7, is displayed in Fig. 1(d), and the resulting map for $w_{\mathrm{TR}}$ on 1 January 2005 is shown in Fig. 2(c). After the initial stratospheric estimate, STREAM yields high values for $T^*$ over

parts of the U.S., Europe, central Africa, and China, resulting in low $w_{\mathrm{TR}}$. The respective satellite pixels will hardly contribute to the stratospheric estimate in the next iteration.

On the other hand, negative $T^*$ over east Canada and Greenland are found in the initial STREAM run, caused by the asymmetric polar vortex. Over the Labrador Sea, initial values for $T^*$ are systematically below the threshold of 0.5 CDU and thus trigger a high $w_{\mathrm{TR}}$, and the respective observations of low total VCDs contribute strongly to the stratospheric estimate in

the next iteration.

### 2.2.4 Total weight

The total weight of each satellite pixel is defined as the product of the individual weighting factors:

$$w_{\mathrm{tot}} := w_{\mathrm{pol}} \times w_{\mathrm{cld}} \times w_{\mathrm{TR}} \tag{8}$$

(i.e., the logarithms as shown in Fig. 2(a)-(c) are simply added, resulting in Fig. 2(d)). The a-priori pollution weight can still

be recognized in the global pattern, but is significantly modified by $w_{\mathrm{TR}}$ (further reducing the overall weight over e.g. the US and China) and $w_{\mathrm{cld}}$, which competes with the pollution weights $<1$. In some regions (e.g. West of the Great Lakes, Scotland, or the Himalayas) the cloud weight shifts the initially low $w_{\mathrm{pol}}$ to a net weight $>1$.

The concept of the combination of different weighting factors is easily extendible by further weights, e.g. based on fire or flash counts in order to account for $NO_x$ emissions from biomass burning or lightning.

### 2.3 Weighted convolution

Global daily maps of the stratospheric column are derived by applying "weighted convolution", i.e. a spatial convolution which takes the individual weights for each satellite pixel into account. This approach is an extension of the "normalized convolution" presented in Knutsson and Westin (1993). Weighted convolution at the same time smoothes and interpolates the stratospheric field. A similar approach was used by Leue et al. (2001), who applied the fitting errors of $NO_2$ SCDs as single weights.

The algorithm is implemented as follows:

- A lat/lon grid is defined with $1°$ resolution. Each satellite pixel is sorted into the matching grid pixel according to its center coordinates. At the $j^{\mathrm{th}}$ latitudinal/$i^{\mathrm{th}}$ longitudinal grid position, there are $K$ OMI pixels with the total columns $V_{ijk}(k=1..K)$ and the weights $w_{ijk}$. We define

$$C_{ij} := \sum w_{ijk} \times V_{ijk} \tag{9}$$

and

$$W_{ij} := \sum w_{ijk} \tag{10}$$





In case of measurement gaps (i.e. $K = 0$), both $C_{ij}$ and $W_{ij}$ are set to 0.

The weighted mean VCD for each grid pixel is then given as

$$V_{ij} = \frac{C_{ij}}{W_{ij}} \tag{11}$$

– A convolution kernel (CK) $G$ is defined (e.g. a 2D Gaussian). Spatial convolution is applied to both $C$ and $W$ (taking the dateline into account appropriately, i.e. $i=1$ and $i=360$ are adjacent grid pixels):

$$\overline{C} := G \otimes C \tag{12}$$

$$\overline{W} := G \otimes W \tag{13}$$

– The smoothed stratospheric VCD for each grid pixel as derived from weighted convolution is then given as

$$\overline{V}_{ij} := \frac{\overline{C}_{ij}}{\overline{W}_{ij}} \tag{14}$$

We illustrate this procedure for a simple 1D example in the supplement (section S2.3 and Fig. S3).

The degree of smoothing is determined by the definition of the CK $G$, which is defined as a 2D Gaussian in STREAM v0.92 with the longitudinal/latitudinal variances $\sigma_\varphi^2$ and $\sigma_\vartheta^2$, respectively. Generally, information on the stratospheric column over polluted regions should be taken from clean measurements at the same latitude. Thus, $\sigma_\varphi$ has to be sufficiently large, while $\sigma_\vartheta$ has to be low as gradients in latitudinal dimension should be mostly conserved. For high latitudes, however, the longitudinal extent of the CK has to be small enough as well in order to be able to resolve the strong gradients caused by the polar vortex.

In order to meet these requirements, we implement the convolution in the following way:

1. Two CKs are defined in order to meet the different requirements for polar vs. equatorial regions:

$$G^{\text{pol}} := G(\sigma_\varphi = 10°, \sigma_\vartheta = 5°)$$
$$G^{\text{eq}} := G(\sigma_\varphi = 50°, \sigma_\vartheta = 10°) \tag{15}$$

   (see Fig. S4 in the supplement).

2. Stratospheric VCDs $V_{\text{strat}}^{\text{eq}}$ and $V_{\text{strat}}^{\text{pol}}$ are derived for both CK according to Eqs. 12- 14.

3. The final stratospheric VCD is defined as the weighted mean of both, depending on latitude $\vartheta$:

$$V_{\text{strat}} := \cos^2(\vartheta) V_{\text{strat}}^{\text{eq}} + \sin^2(\vartheta) V_{\text{strat}}^{\text{pol}} \tag{16}$$

By this method, spatial smoothing is wide enough at the equator (needed to interpolate e.g. the stratosphere over Central Africa), but small enough at the polar vortex.





In latitudinal direction, this procedure can cause small, but systematic biases if stratospheric $NO_2$ shows significant latitudinal gradients on scales of $\sigma_{lat}$ or smaller. To overcome this, STREAM provides the (default) option to run the weighted convolution on "latitude-corrected" VCDs. I.e., the mean dependency of $V^*$ on latitude is 1. determined (again over the Pacific), 2. subtracted from all individual $V_{ijk}$, and 3. added again to the stratospheric estimate from weighted convolution. By this procedure, latitudinal gradients are largely removed for the convolution (but not from the final stratospheric fields), and the systematic biases vanish (as shown in section S2.3).

## 2.4 Data processing

STREAM estimates stratospheric fields and tropospheric residues for individual orbits. For each orbit under investigation, the orbit itself plus the 7 previous and subsequent orbits (corresponding to about $\pm 12$ hours in time, or $\pm 180°$ in space (longitude), for the investigated satellite instruments in polar sun-synchronous orbits) are used for the calculation of $V^*$, weighting factors, and thus $V_{strat}$ via weighted convolution. For the daily means presented in this study, all orbits where the orbit start date matches the day of interest are averaged.

Alternatively, STREAM can be run in "Near-real time" (NRT) mode, in which the 14 past, but no future orbit, are included in the weighted convolution. We discuss the performance of STREAM NRT for the example of GOME-2 in section 5.2.

STREAM v0.92, implemented as MATLAB script at MPI-C, requires about 10 seconds for processing one orbit of OMI data on a normal desktop PC (3.4 GHz). Time consuming steps are, at about equal parts, the sorting of the satellite pixels on the global grid $V_{ijk}$ (see sect. 2.3), and the convolution process, while the time needed for the calculation of weighting factors is negligible.

## 3 Datasets

## 3.1 Satellite datasets

Several UV/vis satellite instruments provide column measurements of atmospheric $NO_2$. Table 1 summarizes the characteristics of the instruments discussed in this study. Below we provide details on the satellite characteristics and the datasets used in this study, starting with OMI (as STREAM was optimized for OMI within TROPOMI verification) and GOME-2, followed by older instrument with particular challenges such as poor spatial coverage (SCIAMACHY) or resolution (GOME).

### 3.1.1 OMI

In this study we mainly focus on OMI for two reasons:

1. OMI provides daily coverage with small ground pixels. While this already results in a high number of available satellite pixels per day ($>10^6$), also the number of clouded pixels matching the requirements to cause a high $w_{cld}$ is high (more than $10^5$ pixels have a $w_{cld} > 5$).





2. STREAM is the STS verification algorithm for TROPOMI. Algorithm testing within TROPOMI verification and comparisons to the TROPOMI prototype algorithm are performed based on actual OMI measurements.

STREAM basically requires $V^*$ (=$S/A_{\mathrm{strat}}$) as input. For OMI, we use the level 2 "OMNO2" data product (version 3) provided by NASA (Bucsela et al., 2013) and labelled as "Standard Product 2" (SP2) therein, which provides de-striped $NO_2$ SCDs and

stratospheric AMFs[2]. In addition, quality proxies are used to exclude dubious measurements (like those affected by the "row anomaly"[3]). Also information on cloud radiance fraction (CRF) and cloud pressure (CP), which is needed for the calculation of $w_{\mathrm{cld}}$, is provided by the OMNO2 v3 hdf files, based on the "improved OMI $O_2$-$O_2$ cloud algorithm" (Bucsela et al., 2013) OMCLDO2.

The NASA v3 product involves a STS algorithm based on a MRSM as well. The resulting tropospheric residues of STREAM

and NASA v3 are compared and discussed in detail in section 5.1.2.

In addition to the NASA product, we also extract the DOMINO (version 2) level 2 data as provided by TEMIS, for two purposes:

1. The TROPOMI "prototype algorithm" (van Geffen et al., 2014) is developed by KNMI based on model assimilation similar to the DOMINO v2 algorithm. Due to the high computational effort of data assimilation, no dedicated TROPOMI verification

data set is available for verification. Instead, we compare the results of STREAM directly to DOMINO v2 (sect. 5.1.1).

2. DOMINO provides TM4 model profiles of $NO_2$ (needed for the calculation of DOMINO tropospheric AMFs). Here, we use the TM4 data in order to construct synthetic total columns of $NO_2$ for performance tests of STREAM (see section 3.3).

Both OMI products are based on the same spectroscopic analysis, i.e. both start with the same $NO_2$ SCD. Note that this SCD is biased high by about 1 CDU due to shortcomings in the spectral retrieval (see van Geffen et al. (2015) and references

therein). Recent algorithm refinements have removed this bias (van Geffen et al., 2015; Marchenko et al., 2015), but updated NASA or TEMIS products are not available yet. However, such an overall bias will be interpreted as stratospheric feature by STREAM and thus does not affect the performance of STREAM with respect to tropospheric residues. The same holds for the operational NASA and TEMIS STS algorithms.

### 3.1.2  GOME-2

The GOME-2 instruments on the Metop-A and -B satellites provide a time-series of almost 10 years with the perspective of continuation until 2025 due to the upcoming instrument on Metop-C. GOME-2 provides a good spatial coverage with moderate satellite ground pixel size.

We applied STREAM to total $NO_2$ columns from the operational product (GDP 4.7), as provided by DLR in the framework of the Ozone Satellite Application Facilities (O3M SAF), for Metop-A.

The operational product uses a MRSM for STS (Valks et al., 2011, 2015) as well. We compare the results of STREAM and the GDP 4.7 algorithm in section 5.2.

---

[2]In the DOMINO v2 product, total SCDs are not de-striped, and stratospheric AMFs are only provided up to a SZA of 80°

[3]http://projects.knmi.nl/omi/research/product/rowanomaly-background.php#overview



### 3.1.3 SCIAMACHY

STREAM was applied to the SCIAMACHY VCDs retrieved at MPIC Mainz (Beirle et al., 2010a; Beirle and Wagner, 2012). While OMI provides daily global coverage, the coverage of SCIAMACHY is rather poor (only about 1/6 of the Earth per day), and ground pixels are larger than for OMI (except for swath edges). Consequently, also the number of total (about 60,000) and cloudy (about 4,000) pixels per day is much lower than for OMI. Thus, SCIAMACHY can be considered as extreme test case for the performance of STREAM.

One reason for the poor spatial coverage of SCIAMACHY is the measurement mode alternating between nadir and limb geometry. This, however, provides the unique SCIAMACHY feature of a direct measurement of the stratospheric column. We thus compare the TR resulting from STREAM to the MPI-C SCIAMACHY product based on LNM (Beirle et al., 2010a), using the MPI-C retrieval scheme for $NO_2$ concentration profiles from limb measurements (Kühl et al., 2008) (sect. 5.3).

### 3.1.4 GOME

GOME was the first nadir-viewing spectrometer in the UV/vis spectral range with a spectral resolution enabling DOAS analyses. Due to large ground pixel size (320 km across-track), only a low number of (total as well as clouded) satellite pixels per day is available. We nevertheless included GOME in this analysis in order to investigate in how far STREAM can be applied within homogenized retrievals for multiple satellite instruments, as planned within the QA4ECV (Quality Assurance for Essential Climate Variables) project[4]. We apply STREAM to the VCDs provided by TEMIS (Boersma and Eskes, 2004) and compare the resulting TRs to a simple RSM (sect. 5.4).

### 3.2 Model data

For comparisons, and for the calculation of synthetic total columns for performance tests of STREAM, we make use of stratospheric $NO_2$ as provided by the ECHAM5/MESSy Atmospheric Chemistry (EMAC) model, which is a modular global climate and chemistry simulation system (Jöckel et al., 2006, 2010, 2015).

We use the results from simulation *RC1SD-base-10a* of the ESCiMo (Earth System Chemistry integrated Modelling) project as detailed by (Jöckel et al., 2015). Here, only basic information on this specific simulation is summarized:

The model results were obtained with ECHAM5 version 5.3.02 (Roeckner et al., 2006) and MESSy version 2.51 at T42L90MA resolution, i.e. with a spherical truncation of T42, corresponding to a quadratic Gaussian grid of approx. $2.8°$ by $2.8°$ in latitude and longitude, and 90 vertical hybrid pressure levels up to 0.01 hPa. The dynamics of the general circulation model was nudged by Newtonian relaxation towards ERA-Interim reanalysis data (Dee et al., 2011).

Simulation *RC1SD-base-10a* was selected from among the various ESCiMo simulations for several reasons: (a) it has been nudged to reproduce the "observed" synoptic situations, (b) its stratospheric resolution is, with $\simeq 65$ levels, finer compared to other simulations from the ESCiMo project, (c) the simulated total column and tropospheric partial column ozone compare well with observations (Jöckel et al., 2015, see), and (d) the precursor emissions from the land transport sector are most realistic in

---

[4]http://www.qa4ecv.eu/





comparison to other simulations. In conclusion, this simulation represents the state-of-the-art in terms of numerical simulation of the atmospheric chemistry. Moreover, the applied nudging technique allows a direct comparison with observational data, since the simulated meteorological situation corresponds to the observed.

Specifically for this study, the submodel SORBIT (Jöckel et al., 2010) was used to extract $NO_2$ mixing ratios along the sun-
synchronous orbit of the Aura satellite, thus matching the local time of OMI observations. Stratospheric VCDs were calculated by vertical integration of the modelled $NO_2$ mixing ratios between the tropopause height (as simulated by EMAC) and the top of the atmosphere.

In this study, we make use of the modelled stratospheric columns for two purposes:

1. We perform a simple model-based STS for comparison. To remove systematic biases between satellite measurements
and EMAC, a latitude dependent offset is determined in the Pacific and corrected for globally, similar as in Richter et al. (2005) and Hilboll et al. (2013). We refer to this EMAC-based STS as $STS_{EMAC}$ and applied it to OMI data (section 5.1.3).

2. Stratospheric VCDs from EMAC are used to construct a synthetic dataset of total $NO_2$ VCDs for performance tests of STREAM (see next section).

### 3.3 Synthetic VCD

We test the performance of STREAM on synthetic VCDs, which allows a quantitative comparison of the estimated TR to the a-priori "truth". The input to STREAM, i.e. synthetic total columns of $NO_2$, should realistically represent (a) stratospheric chemistry and dynamics, (b) tropospheric emissions, transport and chemistry, (c) cloud fields, and (d) the satellite sampling. For these purposes, we construct synthetic $NO_2$ column densities $V^*$ based on
(a) stratospheric VCDs from EMAC[5] at AURA overpass time (sect. 3.2),
(b) modelled tropospheric VCDs from TM4 (sect. 3.1.1), and
(c) measured cloud properties and the respective tropospheric AMFs from OMI as provided in the DOMINO $NO_2$ product.

Synthetic TRs are given as $T^*=V_{trop} \times A_{trop}/A_{strat}$ (compare eq. 4). Synthetic total columns $V^*$ are then calculated as $V_{strat}+T^*$ (eq. 3) and fed into STREAM. The resulting fields of stratospheric VCDs and the respective TRs can then be
compared to the a-priori "truth".

### 4 Algorithm performance

In this section we analyse the performance of STREAM. As the true stratospheric VCD is not known, the error of any STS algorithm is not easily accessible. Still, the STS performance can be evaluated based on the properties of the resulting TR: In remote regions without substantial $NO_x$ emissions, $T^*$ should generally be low, but still positive (about 0.1 CDU, Valks et al.
(2011)). Also the variability of $T^*$ over both space and time should be low in regions free of tropospheric sources.

---

[5]Stratospheric columns are taken from EMAC rather than TM4, as the latter does not represent a free model run of stratospheric chemistry and dynamics, but uses the satellite measurements for assimilation.



Below, we investigate the characteristics of $T^*$ from STREAM (sect. 4.1) and its dependency on a-priori settings (sect. 4.2) for OMI measurements. In addition, the error of $T^*$ is quantified based on synthetic data (sect. 4.3). Application of STREAM to other satellite instruments and the comparisons between STREAM and other STS algorithms are provided in section 5.

## 4.1 Performance of STREAM for OMI compared to RSM

Figure 3 displays the OMI daily mean VCD $V^*$ (top) as well as the respective stratospheric field from STREAM (bottom) for 1st of January (left) and 1st of July 2005 (right), respectively. The overall latitudinal as well as longitudinal dependencies are clearly reflected in the stratospheric fields, while small-scale stratospheric features are lost by the spatial convolution.

Figures 4 and 5 display the tropospheric residues resulting from a simple RSM and STREAM, respectively. Shown are OMI results for January (left) and July (right) 2005 for both daily (top) and monthly (bottom) means. Figure 6 summarizes the daily and monthly statistical properties of TR, i.e. the median as well as $10^{th}/90^{th}$ and $25^{th}/75^{th}$ percentiles (light/dark bars) for different regions (see Figures S6 and S7 for an illustrative sketch of the meaning of the percentile bars, as well as the definition of regions).

Overall, spatial patterns of TR are similar for RSM and STREAM, in particular the enhanced values reflecting tropospheric pollution over e.g. the US, Central Africa, or China. However, RSM reveals several artefacts of both enhanced as well as systematically negative TR as a consequence of the simple assumption of zonal invariability of stratospheric $NO_2$. For instance, on 1 January 2005, VCDs over Northern Canada are lowered due to the polar vortex (Fig. 3 top left). Consequently, the simple RSM results in negative $T^*_{RSM}$ down to -0.7 CDU (Fig. 4). On the other hand, $T^*_{RSM}$ over North-Eastern Russia are quite high ($> 0.5$ CDU). This pattern is slightly reduced, but still present in the monthly mean (see the statistics of $T^*_{RSM}$ for high latitudes in Fig. 6).

This feature is largely reduced by STREAM (Fig. 5 top left). The spread of $T^*$ at high latitudes is more than 3 times lower than that of $T^*_{RSM}$ (Fig. 6). Also for July, systematic structures showing up in $T^*_{RSM}$ (in polar regions, but also in the Indian ocean at 30°-60°S) are largely reduced in STREAM.

Over the Pacific, $T^*_{RSM}$ is, by construction, 0 on average. $T^*$ is systematically higher by about 0.1 CDU (Fig. 6). This results from the emphasis of clouded pixels used for STREAM, which directly reflect the stratospheric rather than the total VCD. This additional advantage of STREAM over RSM is further discussed below (Sect. 5.6).

As both RSM and STREAM generally assume stratospheric patterns of $NO_2$ to be smooth, i.e. do not resolve longitudinal variations at all (RSM) or on scales $< \sigma_\varphi$ (STREAM), the small-scale variations in the total VCD (Fig. 3 top) are transferred to the TR, resulting in "patchy" daily TRs ranging from about -0.1 up to +0.4 CDU in remote regions ($10^{th}$-$90^{th}$ percentiles). In the monthly means, however, these patchy structures have mostly vanished (both for RSM and STREAM), as the spatial patterns of different days at variable locations cancel each other out. The remaining systematic patterns in monthly mean $T^*$ have generally larger spatial scales and are within 0 up to +0.25 CDU in remote regions.

On 1 July, a band of enhanced $V^*$ shows up around 20°-30°S, where (a) $V^*$ is higher in the Indian Ocean compared to the Pacific, and (b) the structure of enhanced $V^*$ is "tilted" in the Pacific (see Fig. 3 top right). I.e., the RSM assumption of zonal invariance is not fulfilled. Consequently, the RSM results in extended horizontal structures ("stripes") of low/high biased





$T^*$ over South America and the Indian Ocean, respectively, ranging from -0.5 up to almost 1.0 CDU (Fig. 4 top right). Again, temporal averaging reduces the amplitude, but systematic patterns of about $\pm$ 0.4 CDU remain in the monthly mean (Fig. 4 bottom right). As STREAM also assumes a weak variation of $V_{\mathrm{strat}}$ with longitude, in particular at low latitudes, the artefacts in $T^*$ are very similar to those of $T^*_{\mathrm{RSM}}$ at 20-30°S. Note that this artefact is particularly strong in July 2005 (as compared to 2010, see section 5.1.4).

In section 5, TRs from STREAM are investigated for other satellite instruments and compared to other STS algorithms, and the advantages and limitations of STREAM are discussed further.

## 4.2   Impact of a-priori settings

STREAM determines the stratospheric $NO_2$ VCD $V_{\mathrm{strat}}$ based on weighting factors as described in Sect. 2. The resulting TRs
thus depend on the weighting factor definition and convolution settings. We performed runs of STREAM with one-by-one modifications of each parameter and compared the results to the baseline setting. Overall, the effects of a-priori settings on $T^*$ have been found to be rather small (of the order of 0.1 CDU), and the STREAM  results are thus robust with respect to the parameters chosen in v0.92.

Below, we summarize the main findings of the performed sensitivity studies. Figures and details are provided in the Supple-
ment.

### 4.2.1   Impact of cloud weight

The cloud weight $w_{\mathrm{cld}}$ was varied

(a) by setting it to 1 (i.e. not accounting for cloud properties at all),

(b) increasing $w_{\mathrm{cld}}$ by a factor of 10,

(c) including high altitude clouds in the calculation of $w_{\mathrm{cld}}$, and

(d) including low altitude clouds in the calculation of $w_{\mathrm{cld}}$.

(a) If no $w_{\mathrm{cld}}$ is applied, the tropospheric estimate over the Pacific is $\approx$0, as for the classical RSM, instead of about 0.1 CDU for the baseline. This difference corresponds to the order of the tropospheric background of $NO_2$. Over polluted regions, however, the difference to the baseline is larger (0.2 CDU). Here, the stratospheric estimate is additionally biased high due to
missing supporting points over continents.

(b) If $w_{\mathrm{cld}}$ is increased by a factor of 10, measurements over clouds by far dominate the stratospheric estimate, yielding lower $V_{\mathrm{strat}}$, and thus higher $T^*$, as compared to the baseline. However, the difference is very small (<0.05 CDU). In addition, the variability of $T^*$ is generally slightly higher in case of a 10 fold increased $w_{\mathrm{cld}}$.

(c) If high altitude clouds are included in the calculation of $w_{\mathrm{cld}}$, the resulting TR hardly changes at all, indicating that the
impact of lightning $NO_x$ on $NO_2$ satellite observations is generally small.

(d) The inclusion of low altitude clouds has almost no effect as well, as expected over clean regions. But over polluted regions, it is expected that low altitude clouds result in increased total columns $V^*$, as soon as there is significant $NO_2$ above or within the cloud, causing high tropospheric AMFs. Consequently, $V_{\mathrm{strat}}$ is expected to be biased high, and $T^*$ biased low



over polluted regions, if low clouds are included in the calculation of $w_{\mathrm{cld}}$. This effect was indeed found, but the absolute change is rather small ($< 0.1$ CDU in winter, almost zero in summer). This weak dependency on the inclusion of low-altitude clouds probably results from the conservative definition of $w_{\mathrm{pol}}$, being already very low over regularly polluted regions.

### 4.2.2 Impact of convolution

In STREAM, two different CK are applied and the final $V_{\mathrm{strat}}$ is calculated as weighted mean of both. We tested the impact of the choice for CK by applying both the polar ("narrow") and equatorial ("wide") CK globally. The narrow CK, and thus the potential range of influence of satellite pixels with high weights, is limited to about $20°$ in longitude. This potentially leads to biases over continents caused by spatial interpolation. Thus, the resulting $T^*$ is (too) low over central Africa. Overall, median $T^*$ over polluted regions is lower compared to the baseline settings by about 0.1 CDU.

For wide CK, on the other hand, the longitudinal gradients at high latitudes are not resolved any more. Consequently, the spatial variability of daily $T^*$ at high latitudes is increased by a factor of 2. We conclude that our choice of the combined CK for high and low latitudes is a good compromise for realising weighted convolution.

### 4.2.3 Impact of latitude correction

If the initial correction of the latitudinal dependency of $V^*$ over the Pacific is omitted, the resulting TR reveals global stripes 15 with negative values around the equator and maxima ($\approx 0.5$ CDU) at about $30°$N/S, both in winter and summer.

### 4.2.4 Impact of the number of considered orbits

In STREAM baseline settings, for each orbit, stratospheric estimation is based on the previous and subsequent 7 orbits, corresponding to full global coverage for OMI. Switching this parameter to either $\pm 14$ or $\pm 3$ orbits has almost no impact on the resulting TR.

In case of NRT application of STREAM, no subsequent orbits are available, and the previous 14 orbits have to be considered. Also this set-up results in essentially the same $T^*$ statistics (compare sect. 5.2).

### 4.2.5 Impact of tropospheric residue weight

In STREAM v0.92, one iteration for $w_{\mathrm{TR}}$ is applied. If $w_{\mathrm{TR}}$ is omitted, the spread of $T^*$ slightly increases for high latitudes. A second iteration does not yield a further improvement. Lowering the threshold in Eq. 7 from 0.5 to 0.3 CDU results in slightly 25 lower spread of $T^*$ at high latitudes in summer.

### 4.3 Performance for synthetic data

In order to estimate the uncertainties of the STREAM stratospheric estimate (and thus tropospheric residues), we apply the algorithm to synthetic input data, as defined in section 3.3, for which the "true" stratospheric fields and TR are known. Again, a simple RSM is applied as well for comparison.





Figure 7 displays the statistics of the error of $T^*$, i.e. the difference $\Delta$ of estimated and a-priori TR, which equals the difference between the true and the estimated stratospheric VCD, for different regions. The spatial patterns of $\Delta$ are shown in the Supplement (Fig. S18).

Over the Pacific, RSM results in TR biased low by 0.1 CDU. With STREAM, the bias is reduced (0.05 CDU), but not completely removed. On 1 Jan 2005, $\Delta$ shows a variability of almost 0.4 CDU ($10^{\mathrm{th}}$ to $90^{\mathrm{th}}$ percentile) for both algorithms. This is mainly caused by the small-scale structures of stratospheric $NO_2$ in EMAC, which are resolved by neither STREAM nor RSM. The respective spatial variability of the monthly mean, however, is much lower (about 0.1 CDU).

Again, the simple RSM results in large biases and high variability of $\Delta$ at high latitudes, which are largely reduced by STREAM.

Overall, the agreement of a-priori and estimated $T^*$ from STREAM is very good, in particular for monthly means. Remaining systematic biases are about 0.1 CDU over polluted regions.

## 5   Comparison to other algorithms and discussion

In this section, we apply STREAM to different satellite instruments, compare the results to various existing STS algorithms, and discuss the challenges, limitations and uncertainties of STS in general and STREAM in detail.

### 5.1   OMI

As shown in the previous section, STREAM as applied to OMI data generally shows a good performance (Figs. 5 and 6). The systematic artefacts of a simple RSM, such as the large variability of $T^*$ at high latitudes, are largely removed by STREAM. In addition, the application of $w_{\mathrm{cld}}$ emphasizes cloudy observations which directly reflect the stratospheric column. Mean $T^*$ over the Pacific is thus not 0 any more as in RSM, and an additional correction for the tropospheric background is not required in STREAM.

The sensitivity of STREAM on a-priori parameters has been found to be small. Remaining monthly mean TRs in clean regions and their variability are of the order of 0.1 CDU.

Below, we compare the OMI results for 2005 to other algorithms, i.e. the operational DOMINO (sect. 5.1.1) and NASA (sect. 5.1.2) data products as well as a simple model-based correction using EMAC (sect. 5.1.3). Figure 8 summarizes the statistics of regional $T^*$ from the different algorithms. Note that only coincident measurements where all 4 data products exist are included in Fig. 8 in order to allow for a meaningfull comparison; in particular, high latitudes in hemispheric winter are skipped, as DOMINO data is not provided for SZA>80°. Thus, the statistics for STREAM slightly differ from those shown in Fig. 6.





### 5.1.1 Comparison to DOMINO

STREAM is part of the TROPOMI verification activities. The operational TROPOMI ("prototype") algorithm for STS of $NO_2$ (van Geffen et al., 2014) was developed by KNMI, based on the DOMINO data processor for OMI (Boersma et al., 2011). The STS therein is done by assimilating the satellite measurements in the CTM TM4 (Dirksen et al., 2011).

For TROPOMI verification, we compare STREAM results for OMI to the respective DOMINO product as shown in Fig. S19. On daily basis, "patchy" patterns of enhanced as well as negative TR show up over remote regions (Fig. S19), which result from the dynamical features already present in total VCDs (Fig. 3) combined with the respective dynamics prognosed by the model; spatial mismatch of these patterns can easily cause biases of the estimated TR in both directions. Interestingly, some patterns look even reversed as compared to STREAM (Fig. 5), for instance South-East from South Africa (around 50°S,

50°E). In the monthly means, these patches again are mostly cancelled out.

Mean regional TR (Fig. 8) are very similar between STREAM and DOMINO. However, the variability of $T^*$ is slightly higher for DOMINO, in particular at high latitudes, but also in the Pacific and in remote regions in July.

Figure 9 displays the differences of the monthly mean TR for January and July 2005. Overall, the differences are quite small (below $\pm0.1$ CDU for 65% of the world between 60°S and 60°N). But still, the monthly means reveal systematic regional

deviations of more than $\pm0.3$ CDU (for less than 3% of the world).

In January, TRs over East Asia at high latitudes are systematically higher for STREAM. This is probably related to an underestimation of DOMINO, as the DOMINO TRs are very low and partly negative in this region. Over North America, TR from STREAM are higher than from DOMINO at the East Coast, but vice versa over Western Canada. In both cases, the lower TR is slightly negative, indicating an overestimation of $V_{\mathrm{strat}}$ from DOMINO/STREAM at the East/West coast, respectively.

In July, the "stripe"-like structure in STREAM has been already discussed in section 4.1. In DOMINO, similar bands of enhanced tropospheric residue are found around 30°S, in particular in the Indian ocean. As the amplitude and width of these bands is different for STREAM and DOMINO, this feature is most striking in the difference map; TRs around 30°S are generally higher for DOMINO.

DOMINO reveals some patches of systematically enhanced TR which are not observed by STREAM and thus show up in

the difference map as well (west of the US, west of the Sahara, Himalaya). Reasons for these regionally enhanced TR (and thus low biased stratosphere) have to be investigated in future studies.

### 5.1.2 Comparison to NASA

The official OMI $NO_2$ product provided by NASA uses a MRSM for STS as well, as decribed in Bucsela et al. (2013). Daily and monthly maps of TR from NASA (OMNO2 v3/SP2) are shown in Fig. S20.

The NASA STS corrects for the tropospheric background based on a "fixed model estimate" (Bucsela et al., 2013). Consequently, TR are about 0.1 to 0.3 CDU over clean regions throughout the world.

TR from NASA are impressively smooth even on daily basis. This results from the STS algorithm which, over clean regions, interprets the full total column as stratospheric column, whenever the quotient of the modelled tropospheric slant column and





stratospheric AMF (matching our definition of $T^*$) goes below a threshold of 0.3 CDU. Thus, at Southern high latitudes in July (completely classified as unpolluted in Bucsela), the TR is almost 0±0, i.e. shows no variability at all (compare Fig. 8) just by construction, as all the variability present in the total column was assigned to the stratospheric column.

While this is probably a reasonable procedure over completely clean regions, we would like to point out that:

1. The smoothness of NASA TR over oceans is not surprising, as it is reached by construction. In particular, the smooth patterns for TR over oceans allow no conclusion on the NASA STS performance over polluted continental regions, where TR are based on interpolated stratospheric fields, just as in STREAM.

   2. The NASA procedure of assigning the total column variability in clean regions completely to the stratospheric estimate also removes any cloud dependency of the TR, which affects applications such as profile retrievals by cloud slicing (e.g.,
Belmonte Rivas et al., 2014).

   3. The NASA procedure runs the risk of labelling episodical $NO_2$ transport events over oceans as stratospheric pattern. Bucsela et al. (2013) perform an automatic "hot spot" identification and elimination scheme to avoid this. But, still, on 1st of January, a $NO_2$ transport event can be seen in the total VCD East of Canada (Fig. 3 top left) which is similar to the "meteorological bomb" described in Stohl et al. (2003). This event is clearly visible in $T^*$ from STREAM (Fig. 5 top
left), but only weakly in $T^*$ from NASA (Fig. S20 top left). The reason for this discrepancy is that the local enhancement of $NO_2$ is partly classified as a stratospheric feature in the NASA product, as illustrated in Fig. S21.

Figure 10 displays the differences of the monthly mean TR for January and July 2005. Again, overall agreement is very good: In January, both products agree within 0.1 CDU for 69% of the Earth, and within 0.3 CDU anywhere. In July, agreement within 0.1/0.3 CDU is found for 64%/94% of the Earth (for latitudes below 60°), respectively. Again, the band at 30°S sticks out
in the difference map as discussed above. Highest deviations of up to 0.5 CDU, however, are observed over the Sahara. Within the NASA STS, the Sahara is masked out completely, as the high albedo and low cloud fractions result in high tropospheric AMFs, such that even low tropospheric VCDs could contribute significantly to the total column. In STREAM, however, large parts of the Sahara are treated as unpolluted and are assigned with $w$=1. A close check of the stratospheric estimates from STREAM and NASA over the Sahara reveals that the large deviation probably results from both, a high biased $V_{\mathrm{strat}}$ by
STREAM, and a low biased $V_{\mathrm{strat}}$ by NASA (see Fig. S22).

### 5.1.3   Comparison to $STS_{EMAC}$

We have used the stratospheric 3D mixing ratios provided by EMAC in order to perform a simple model-based STS, similar as in Hilboll et al. (2013). First, the latitude-dependent offset between EMAC and OMI VCDs is estimated over the Pacific. Second, the offset-corrected stratospheric $NO_2$ VCDs is used for global STS. No additional correction for the tropospheric
background is performed, such that the mean TR over the Pacific is 0 by construction.

Daily and monthly maps of TR from $STS_{EMAC}$ are shown in Fig. S23. Daily maps reveal patches of TR from -0.3 CDU up to 0.4 CDU resulting from mismatches in actual and modelled stratospheric dynamics. In the monthly mean, these fluctuations



cancel out at large part. Overall, variability ($10^{th}$-$90^{th}$ percentiles) of $T^*$ in remote regions was found to be about 0.3-0.4, similar as for DOMINO.

Figure 11 displays the differences of the monthly mean TR for January and July 2005. The overall negative values over ocean are a result of the neglect of the tropospheric background in $STS_{EMAC}$. Besides this, most striking features are

1. positive deviations (i.e., TR from $STS_{EMAC}$ being higher than from STREAM) over North America and Eurasia in January (up to 0.45 CDU, North from 35°N),

2. negative deviations over North America and Eurasia in July (down to -0.45 CDU, North from 35°N), and

3. positive deviations over the Sahara, Middle East, India and Western China in July (up to 0.38 CDU).

The systematic deviations North from 35°N (1.&2.) are caused by the longitudinal dependency of $T^*$ from $STS_{EMAC}$ (see Fig. S23) and indicate that the mean longitudinal dependency of stratospheric $NO_2$ is not fully reproduced by EMAC. Deviations in July over Sahara and Southern Asia (3.), however, are at least partly caused by a low bias of $T^*$ from STREAM as discussed in the previous section.

Overall, deviations are moderate, and $STS_{EMAC}$ still improves the statistics of TR for high latitudes as compared to a simple RSM. It thus might be considered as a simple alternative STS with the advantage that it can be expected to work with the same performance for any satellite instrument, independent on spatio-temporal coverage.

### 5.1.4 OMI after row anomaly

In 2005, OMI measurements have been performed with good instrumental performance, providing daily global coverage. This has changed since summer 2007, when radiance measurements of poor quality regularly occurred at particular cross-track positions ("row anomaly"). We thus also tested STREAM on OMI data after onset of the row anomaly: Figures S24 and S25 in the supplement show $T^*$ for 2010. While the daily maps reveal gaps due to the exclusion of measurements affected by the row anomaly, the monthly mean patterns as well as the statistical properties are comparable to the results for 2005. The row anomaly thus does not impact the performance of STREAM (nor DOMINO and NASA retrievals).

### 5.2 GOME-2

STREAM has been applied to GOME-2 (Metop-A) data for the year 2010. The resulting daily and monthly mean maps are shown in Fig. S26. Again, statistical properties are summarized in Fig. 12.

The overall performance of STREAM, i.e. median and variability range of TR, are generally similar to that found for OMI. However, while OMI TR are about 0.1 CDU over the Pacific, lower values (0.05 CDU) are found for GOME-2. This might be related to the lower number of clouded pixels for GOME-2, differences in the cloud products, or systematic spectral interferences caused by clouds in either algorithm.

On 1st of July 2010, GOME-2 is operated in narrow swath mode, causing poor global coverage. This however does not affect STREAM performance.





On 15 January 2010, STREAM results in extraordinary high TR over the ocean (Fig. S27), which turned out to be caused by a solar eclipse. Removing the affected orbit results in normal performance for this day. We recommend that screening of solar eclipses is done automatically (as done for OMI) before running any STS algorithm.

### 5.2.1 Comparison to Near Real Time (NRT) mode

STREAM is foreseen to be implemented in an update of the operational GOME-2 data processor as operated in the framework of the O3M SAF. This requires a slight modification of STREAM in order to work on NRT data.

In STREAM v0.92, the stratospheric fields are estimated for each orbit based on the total column measurements including 7 previous and 7 subsequent orbits. In NRT, however, no subsequent orbits are available. Thus, STREAM has to be operated on the current plus 14 previous orbits instead.

We ran STREAM in NRT mode. The resulting maps are shown in Fig. S28, and the statistics of TR are included in Fig. 12. The deviations between baseline and NRT are marginal. Thus, STREAM can be operated in NRT with stable performance.

### 5.2.2 Comparison to operational product (GDP 4.7)

In the current operational data processor (GDP 4.7), STS for $NO_2$ is done by a MRSM as described in Valks et al. (2011, 2015). Basically, polluted regions (defined by monthly mean TVCDs from the MOZART-2 model being larger than 1 CDU)
are masked out. Global stratospheric fields are derived by low pass filtering in zonal direction by a $30°$ boxcar filter.

Figure S29 displays daily and monthly mean maps of $T^*$ in January and July 2010. The respective regional statistics are included in Fig. 12.

Overall, TRs from GDP are relatively low. Over the Pacific, mean $T^*$ is close to 0 in January, despite the applied tropospheric background correction of 0.1 CDU. Over polluted regions, median TR from GDP is systematically lower (by 0.2 CDU in July)
than from STREAM, and almost a quarter of all TRs are even negative.

Figure 13 displays the differences of the monthly mean TR from GDP 4.7 and STREAM for January and July 2010, again pointing out the systematically lower values of GDP TR over continents in July.

The systematic low bias of GDP TR probably results from moderately polluted pixels over regions labelled as "unpolluted", but can still reach TVCDs up to 1 CDU in MOZART. These measurements cause a high bias of the estimated stratospheric
field around polluted regions; by the subsequent low-pass filtering, this high bias is passed over the polluted regions and results in low-biased TR. Further investigations are needed to find out why this effect is stronger in July than in January.

### 5.3 SCIAMACHY

We have applied STREAM to SCIAMACHY VCDs from the MPI-C $NO_2$ retrieval (Beirle et al., 2010a). The resulting daily and monthly mean maps for 2010 are shown in Fig. S30. Regional statistics are provided in Fig. 14, again compared to the
simple RSM, and, in addition, to the results of limb-nadir-matching (LNM).





Though SCIAMACHY provides poorer daily spatial coverage, STREAM overall works still well. Again, a clear reduction of the variability of $T^*$ is found at high latitudes as compared to RSM. Over the Pacific, mean TR from STREAM is higher than for the RSM (=0), but, similar as for GOME-2, not as high as for OMI. Again, this could be related to the low number of cloudy satellite pixels and spectral interferences, affecting the DOAS analysis, related to clouds. Overall, regional statistics of

$T^*$ are similar as for OMI or GOME-2. However, a systematic latitudinal dependency of $T^*$ remains, showing positive values in hemispheric summer and negative values in hemispheric winter. This results from the latitudinal dependencies of $V^*$ being different for clouded and cloud free observations, as shown in Fig. S32, for reasons not yet understood.

### 5.3.1  Comparison to LNM

Within the SCIAMACHY MPIC $NO_2$ retrieval (Beirle et al., 2010a), STS is performed based on LNM. This can be considered

as a completely different STS approach, based on actual measurements, but not involving CTMs or large-scale interpolation, and thus provide a valuable information on STREAM performance. Figure S31 displays the daily and monthly mean TR from LNM in 2010.

In the LNM STS, the latitude-dependent offset between nadir and limb is determined over the Pacific and corrected for globally. I.e., mean TR in the Pacific is 0 by construction. Overall, regional statistics of $T^*$ from LNM are very similar to those

from STREAM. Fig. 15 displays the monthly mean difference of both algorithms. The deviation is dominated by the latitudinal dependency of $T^*$ from STREAM (see above); if this is removed, both algorithms agree within 0.2 CDU for most parts of the globe.

### 5.4  GOME

GOME was the first instrument of the investigated series of UV/vis spectrometers suited for DOAS analyses of tropospheric

trace gases. The comparably small number of GOME pixels and the large across-track footprint (320 km) required a modification of STREAM: We have switched the resolution of the global grid used for weighted convolution from 1° to 5°, i.e. wider than the GOME across-track width at moderate latitudes. Thereby it is ensured that a grid pixel usually contains multiple satellite pixels, and that also adjacent grid pixels are not empty (as would be the case for an 1° grid), which is the prerequisite for the calculation of $w_{TR}$.

We have applied STREAM to GOME data as provided by TEMIS. The resulting TRs are again compared to the simple RSM. Figure 16 displays the regional TR statistics for GOME in January and July 1999. The respective maps are provided in the Supplement (Fig. S33).

Overall, STREAM yields reasonable results for GOME as well. However, some systematic biases are observed:

- over the Pacific, TR from STREAM were found to be negative. This can only be explained if the measured columns for

cloudy pixels are higher than for cloud free pixels.

- over polluted regions, $T^*$ from STREAM is systematically lower than from RSM (by 0.2 CDU in July). This might as well be a consequence of the applied cloud weight, which has obviously different effects on GOME than on OMI.





This explanation would be consistent with previous findings: while Leue et al. (2001) base the STS on cloudy pixels, Wenig et al. (2004) switched the Heidelberg STS to cloud-free pixels, after noticing that GOME columns are higher instead of lower over clouds. Wenig et al. (2004) relate this to the contribution of lightning $NO_x$. However, as (a) the impact of lightning $NO_x$ on satellite observations is generally small (Beirle et al., 2010b), and (b) lightning activity over the remote Pacific used for the RSM is very weak, we rather suspect that a different effect is responsible for this finding, most probably related to the specific instrumental features of GOME (Burrows et al., 1999), in particular the dichroic mirrors causing polarization dependent spectral structures. It might thus be worth re-checking the DOAS analysis of $NO_2$ for GOME for spectral interferences related to clouds. A second possible effect, which might in particular contribute to the large discrepancy over polluted regions, is that cloud properties are averaged over the large GOME ground pixel. I.e., in an extreme case, low and high cloud layers, which would both be skipped in $w_{cld}$ if resolved by the satellite pixel, might yield, on average, an effective cloud height yielding a high $w_{cld}$. Any tropospheric pollution within (or directly above) the low cloud layer would then bias high the stratospheric estimate, and bias low the TR.

## 5.5 Future instruments

### 5.5.1 TROPOMI

TROPOMI (Veefkind et al., 2012) on Sentinel 5p will be launched in 2016. Instrumental set-up and spatial coverage are similar to OMI, but TROPOMI will provide a better spatial resolution of $7{\times}7$ km$^2$.

STREAM was developed as verification algorithm for TROPOMI STS and was tested and compared to the TROPOMI prototype algorithm based on OMI measurements (see above). Though no TROPOMI measurements are available yet, it can be expected that the performance of STREAM on TROPOMI will be even better than for OMI, since due to the better spatial resolution, more individual satellite pixels are available, and among them a higher fraction of clouded pixels. Thus, more sampling points over potentially polluted regions will be available, further decreasing interpolation errors.

### 5.5.2 Sentinel 4

The satellite instruments investigated so far are all operated in low, sun-synchronous orbits, providing global coverage at fixed local time. In the near future, a new generation of spectrometers on geostationary orbits will be launched by different space agencies. Over Europe, Sentinel 4 (S4, Ingmann et al., 2012) will be the first geostationary spectral resolving UV/vis instrument. The spatial coverage is focussed on Europe. Thus, no "clean" reference regions are regularly available. STREAM might overcome this problem by using clouded observations where the tropospheric pollution is effectively shielded.

We simply evaluate the expected performance of STREAM on S4 measurements by clipping OMI measurements to the area covered by S4 (as given in Courrèges-Lacoste et al., 2010). The STREAM settings are identical to v0.92, except for the a-priori removal of the overall latitude dependency in the reference sector, as no Pacific measurements are available for S4. Fig. 17 displays the resulting TR (top) and the difference of TR between clipped and global OMI data (bottom) for January 2005.



Though tropospheric pollution over Europe and the Middle East is evident, i.e. an extended clean reference region is actually not available, STREAM is still capable of yielding an accurate stratospheric estimate. Only at the northern and southern borders, systematic biases are observed, which can be caused by the overall latitudinal dependency of the stratospheric VCD and border effects of the weighted convolution, and can probably be reduced by dedicated optimization of the algorithm for S4.

Situation will probably be improved for real S4 measurements due to the higher number of clouded pixels in S4 compared to OMI. Thus, this first check is highly encouraging to further investigate the applicability of STREAM to S4 and possible improvements.

## 5.6  Advantages and limitations of STREAM

STREAM was successfully applied to various satellite measurements with a wide range of spatial resolution and coverage.
STREAM is a MRSM and does not need any model input. It can thus be considered as a complementary approach to data assimilation, as chosen for the TROPOMI prototype algorithm.

As (M)RSMs usually estimate the stratospheric column based on total column measurements over clean regions, they generally miss the (small) tropospheric background of the order of 0.1 CDU. Several (M)RSMs explicitly correct for this effect based on a-priori tropospheric background columns (Martin et al., 2002; Valks et al., 2011; Bucsela et al., 2013). In case of
STREAM, however, cloudy pixels, which allow a direct measurement of the actual stratospheric column (except for the small troposheric column above the cloud), are emphasized. Thus, an additional tropospheric background correction is not required. Accordingly, in case of OMI, TR from STREAM are about 0.1 CDU over clean regions, similar as for TR from DOMINO and NASA.

In case of other satellite instruments, however, the TR over the Pacific was found to be lower (GOME-2 and SCIAMACHY)
or even negative (GOME-1). The latter can only be explained by cloudy measurements being systematically higher than cloud-free measurements. Further investigations are needed to infer this discrepancy between OMI and GOME-1/2/SCIAMACHY, and how far it is related to differences in the cloud products and/or the spectral analysis of $NO_2$.

STREAM assumes stratospheric $NO_2$ fields having low zonal variability, in particular at low latitudes. This is reflected by the choice of a wide convolution kernel at the equator. STREAM is thus not capable of resolving diurnal small-scale patterns
caused by stratospheric dynamics. These patterns, however, at large part cancel out in monthly means.

Whenever actual stratospheric fields do not match the a-priori assumption of zonal smoothness, e.g. in case of "tilted" structures or actual large-scale zonal gradients, like differences in the stratospheric column over Pacific and Indian ocean, the TR resulting from STREAM can show artificial "stripes". Further investigations might lead to additional sophisticated algorithm steps to remove these artefacts. However, it has to be taken care that the benefit really outbalances the drawbacks
(added complexity), and that no other artefacts/biases are introduced.

The dependencies of TR on STREAM parameter settings have been found to be low ($\lesssim 0.1$ CDU). The application of STREAM on synthetic data results in deviations to the a-priori truth of the same order. These deviations are systematic, i.e. the stratospheric patterns estimated by STREAM are slightly biased high, which can be expected, as they are based on total column measurements, which are always higher than the stratospheric column.





Overall, STREAM uncertainty is well within the general uncertainties of STS (see next section). Note that systematic changes of the $NO_2$ columns of the same order of 0.1 CDU can also result from changes of the settings for the DOAS analysis, like fit interval, inclusion of additional absorbers in the analysis, or the treatment of rotational and vibrational Raman scattering, creating overall biases as well as spatial patterns, e.g. over oligotrophic oceans (Enno Peters, personal communication).

## 5.7 General uncertainties and challenges of Stratosphere-Troposphere Separation

The uncertainty of STS can often not be directly quantified, as the "true" stratospheric 4D concentration fields are not known. But still, the TR can be used to evaluate the plausibility of the stratospheric estimate and to derive realistic uncertainties:

- Negative TRs are nonphysical. Thus, the occurence of negative $T^*$ (exceeding the values/frequencies explainable by noise) clearly indicates a positive bias of the estimated stratosphere.

- Tropospheric background columns over regions free from $NO_x$ sources are expected to have low spatio-temporal variability. Thus, the observed variability of $T^*$ over clean regions serves as proxy of the uncertainty (precision) of the STS. From different algorithms (MRSMs as well as model-based methods), typical variabilities of $T^*$ over remote regions are about 0.5 CDU for daily means, and about 0.2-0.3 CDU for monthly means. For a simple RSM, much higher values ($\approx$ 1 CDU) are found at high latitudes.

Systematic biases (accuracy) of STS can be estimated from the inter-comparison of TRs from different algorithms. Fig. 18 displays the standard deviation of monthly mean TR from the algorithms shown in Fig. 8 and discussed in Sect. 5.1, i.e. two different, independent MRSM approaches (STREAM and NASA) as well as two STS based on models, a simple one ($STS_{EMAC}$) and a complex data assimilation set-up (DOMINO). Note that the upper range of the colorbar was lowered to 0.3 CDU.

Overall, the standard deviation of TR from different STS is low (typically $<0.1$ CDU, and below $<0.2$ CDU for most parts of the world). It is thus consistent with the uncertainty estimates of stratospheric columns given in literature (Boersma et al. (2011): 0.15-0.25 CDU (SCD); Valks et al. (2011): 0.15-0.3 CDU (VCD); Bucsela et al. (2013): 0.2 CDU (VCD)).

With respect to the final $NO_2$ TVCD product, which is higher than TR by the ratio of stratospheric and tropospheric AMFs (Eq. 4), uncertainties of this order are completely negligible over polluted regions such as the US Eastcoast, central Europe, or Eastern China. But still, a regional bias $>0.2$ CDU (e.g. over Russia in January) can contribute significantly to the relative uncertainty of TVCDs aside the pollution hotspots. Thus, the uncertainty of STS has to be kept in mind in studies focussing on $NO_x$ emissions from e.g. biomass burning or soil emissions over regions like Siberia, the Sahel, or Australia.

## 5.8 Other trace gases

STREAM was developed as STS algorithm for $NO_2$. However, several other trace gas satellite retrievals face problems which are similar to STS from an algorithmical point of view, i.e. that a small-scale tropospheric signal has to be separated from a smooth background (e.g. caused by stratospheric columns, or, in particual in case of trace gases with low optical depth, shortcomings of the spectral analysis, introducing artificial dependencies on e.g. SZA or ozone columns). Thus, modifications of STREAM could be used within the satellite retrievals of e.g. $SO_2$, BrO, HCHO, or CHOCHO.



## 6 Conclusions

The separation of the stratospheric and tropospheric column is a key step in the retrieval of $NO_2$ TVCDs from total column satellite measurements. As coincident direct measurements of the stratospheric column are usually not available (except for SCIAMACHY), current STS algorithms either use CTMs (directly or via data assimilation) or follow a modified reference
sector method (MRSM) approach, where the stratospheric columns are basically estimated from total column measurements over clean regions.

We have developed the MRSM STREAM. Weighting factors determine how far individual satellite pixels contribute to the stratospheric estimate. Over potentially polluted regions (according to an $NO_2$ climatology), weights are lowered, whereas measurements over mid-altitude clouds are assigned with a heigh weighting factor. Global stratospheric fields are derived by
weighted convolution, and subtracted from total columns to yield tropospheric residues (TR). In a second iteration, weighting factors are modified based on the TR: high TR indicates tropospheric pollution, and the respective satellite pixels are assigned with a lower weight. For systematically negative TR, on the other hand, weighting factors are increased. The concept of multiplicative weights can easily be extended by additional factors, e.g. based on fire counts in order to explicitly exclude biomass burning events.

STREAM results are robust with respect to variations of the algorithm settings and parameters. With the baseline settings, the errors of STREAM on a synthetic dataset have been found to be below 0.1 CDU on average.

STREAM was successfully applied to satellite measurements from GOME 1/2, SCIAMACHY, and OMI. The resulting TR over clean regions and their variability have been found to be low. However, systematic "stripes" can appear in STREAM TR if the basic assumption that the stratospheric column varies smoothly with longitude is not given, e.g. in case of "tilted"
stratospheric patterns.

The emphasis of clouded observations, which provide a direct measurement of the stratospheric rather than the total column, should supersede an additional correction for the tropospheric background, which successfully worked for OMI, but less for GOME and SCIAMACHY, for reasons not yet fully understood.

STREAM, which was developed as TROPOMI verification algorithm, was optimized for OMI measurements. Within an
O3M SAF visiting scientist project, it was also applied to GOME-2, and STREAM is foreseen to be implemented in an upcoming GDP update.

Results from STREAM were compared to the TROPOMI prototype algorithm, as represented by the DOMINO v2 product, in which STS is implemented by data assimilation. Differences between monthly mean TR from STREAM and DOMINO are found to be low (almost 0 on average with regional patterns up to about ±0.1-0.2 CDU). Comparison to other state-of-the-art
STS schemes yields deviations of similar order.

The impact of STS is thus generally negligible for TVCDs over heavily polluted regions. But the remaining uncertainties still contribute significantly to the total error of TVCDs over moderately polluted regions and have to be kept in mind for emission estimates of area sources of $NO_x$ such as soil emissions or biomass burning.





*Acknowledgements.* We would like to thank the agencies providing the satellite data: The OMI OMNO2 product is archived and distributed from the Goddard Earth Sciences Data & Information Services center (NASA). We acknowledge the free use of GOME and OMI (DOMINO) level2 data from www.temis.nl. GOME-2 GDP 4.7 level2 data is provided by DLR within the framework of EUMETSAT's O3M SAF. SCIAMACHY level1 data is provided by ESA.

5    This study was supported by the following projects: Within TROPOMI verification, funding was provided by DLR Bonn under contract 50EE1247 for the development of STREAM and the comparison to the TROPOMI prototype algorithm. Within an O3M SAF visiting scientist project, STREAM was adopted to GOME-2 and compared to the current GDP 4.7 data product. Steffen Beirle acknowledges funding from the FP7 Project QualityAssurance for Essential Climate Variables (QA4ECV), n° 607405.

    We acknowledge fruitful discussions with and valuable feedback from many colleagues, in particular during the TROPOMI verification
10   and QA4ECV meetings, especially Andreas Richter, Andreas Hilboll (both IUP Bremen, Germany), Folkert Boersma, and Henk Eskes (both KNMI De Bilt, The Netherlands). Andreas Stohl (NILU, Kjeller, Norway) is acknowledged for discussions on $NO_2$ transport.



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



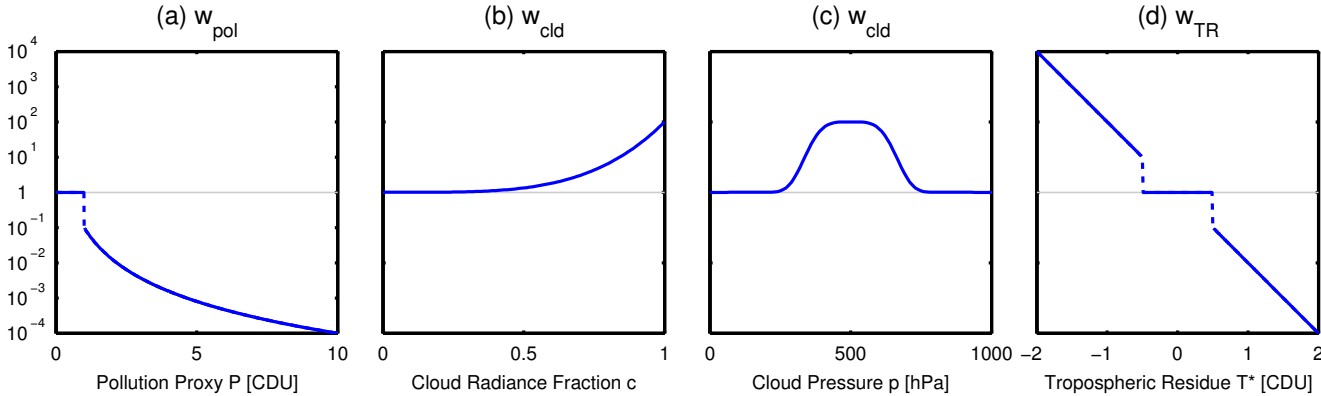

**Figure 1.** Definition of weighting factors (a) $w_{pol}$ as function of the pollution proxy $P$ (Eq. 5), (b) $w_{cld}$ as function of the cloud radiance fraction (Eq. 6) for a cloud pressure of 500 hPa, (c) $w_{cld}$ as function of the cloud pressure (Eq. 6) for a cloud radiance fraction of 1, and (d) $w_{TR}$ as function of the tropospheric residue (Eq. 7).





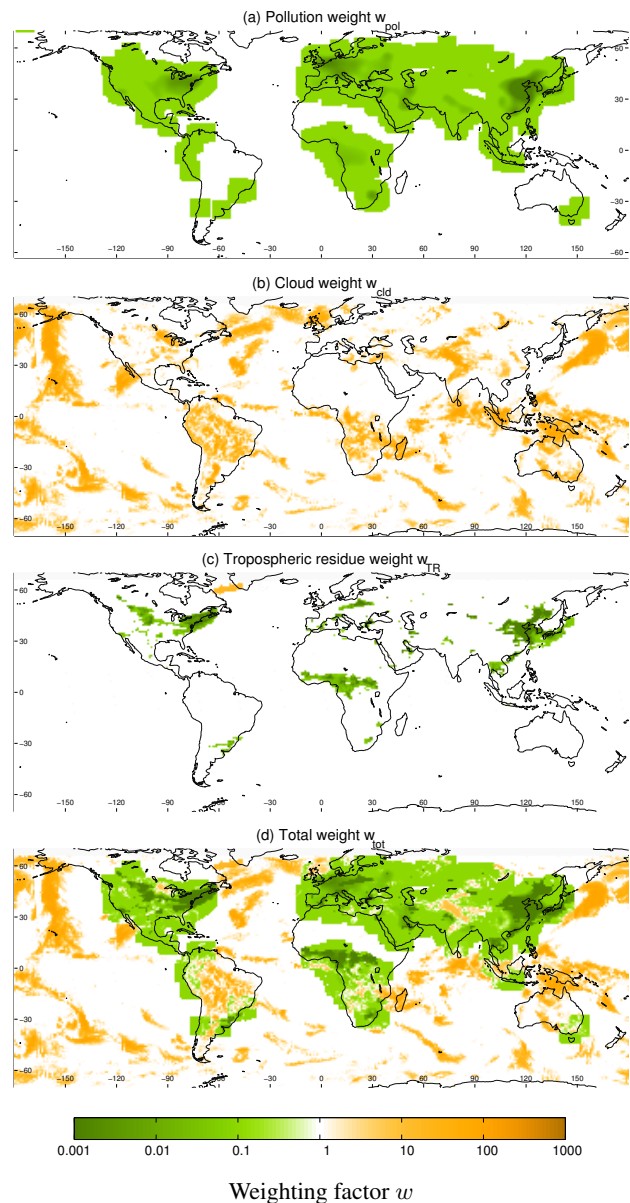

**Figure 2.** Maps of the weighting factors for 1 January 2005 for OMI. (a) Pollution weight $w_{\mathrm{pol}}$. (b) Cloud weight $w_{\mathrm{cld}}$. (c) Tropospheric residue weight $w_{\mathrm{TR}}$. (d) Product of all weighting factors (Eq. 8).



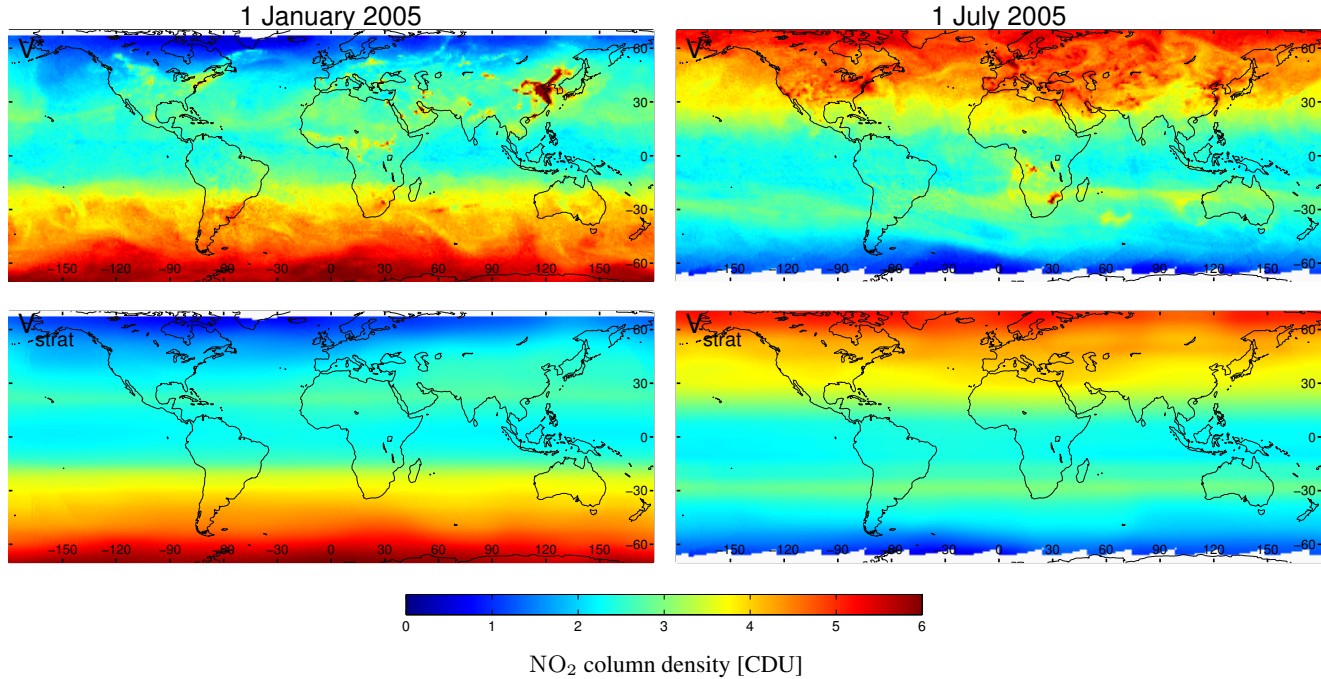

**Figure 3.** Total OMI VCD $V^*$ (top) and the resulting stratospheric estimate $V_{\mathrm{strat}}$ from STREAM (bottom) for 1 January (left) and 1 July (right) 2005.





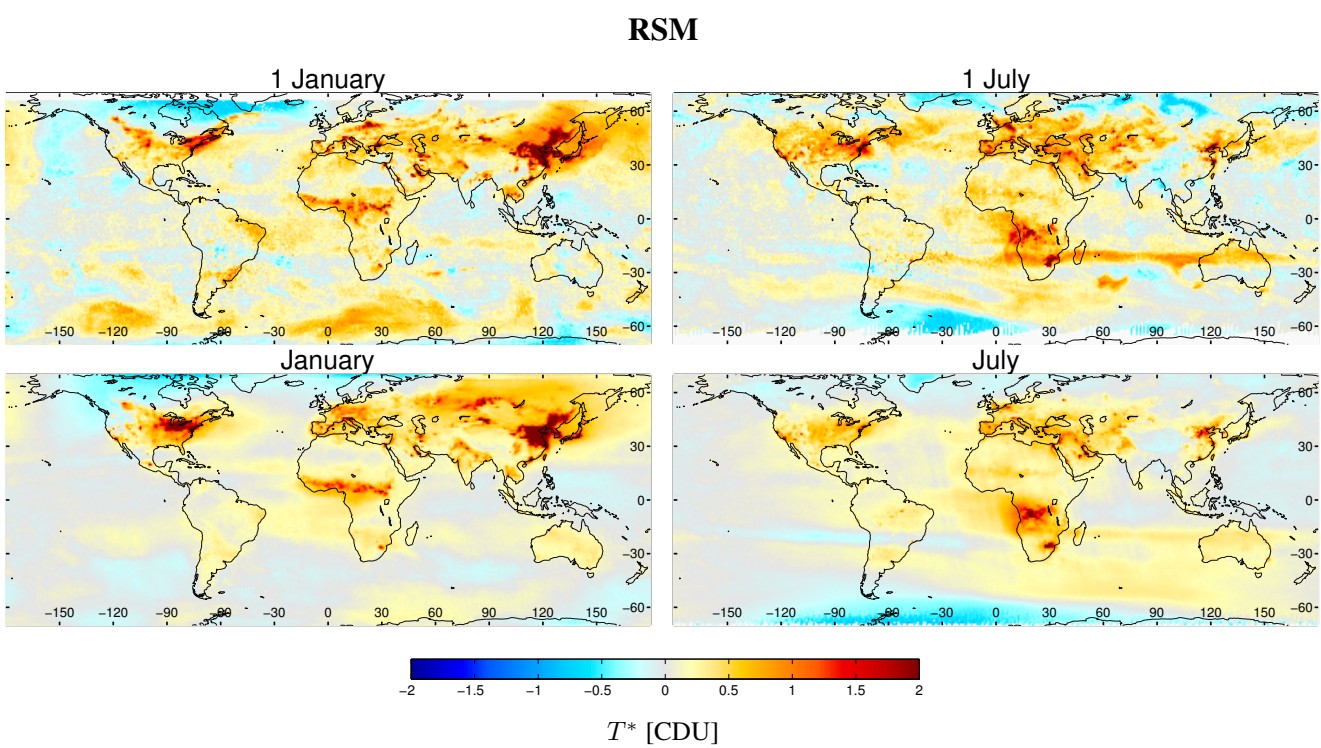

**Figure 4.** OMI tropospheric residues $T^*$ based on RSM for January (left) and July (right) 2005 for the first day of the month (top) and the monthly mean (bottom).





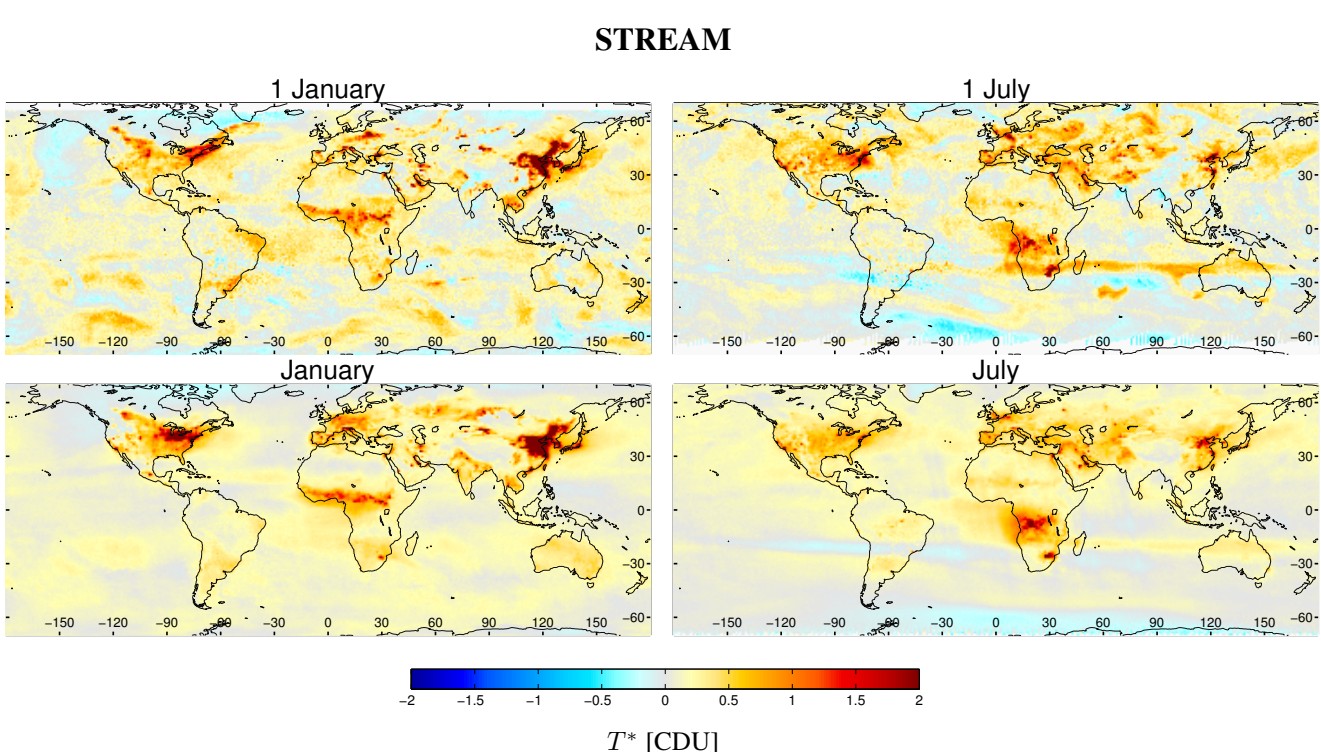

**Figure 5.** OMI tropospheric residues $T^*$ based on STREAM for January (left) and July (right) 2005 for the first day of the month (top) and the monthly mean (bottom).





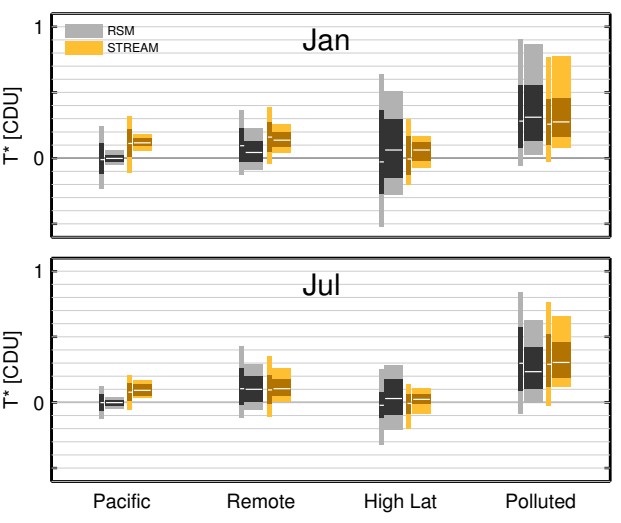

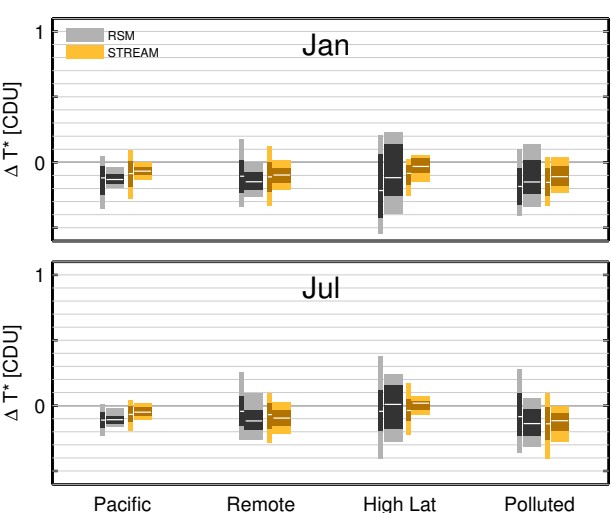

**Figure 6.** Regional statistics of OMI tropospheric residues $T^*$ from RSM and STREAM for January (top) and July (bottom) 2005. Light and dark bars reflect the 10-90[th] and 25-75[th] percentiles, respectively. The median is indicated in white. Narrow bars show the statistics for the first day of the month, wide bars those of the monthly means (see also Fig. S6 for illustration). The regions are defined in Fig. S7. "High latitudes" refer to the respective hemispheric winter only.

**Figure 7.** Regional statistics of the error of $T^*$ from STREAM, i.e. the difference of estimated and a-priori TR (based on synthetic total columns as defined in section 3.3).




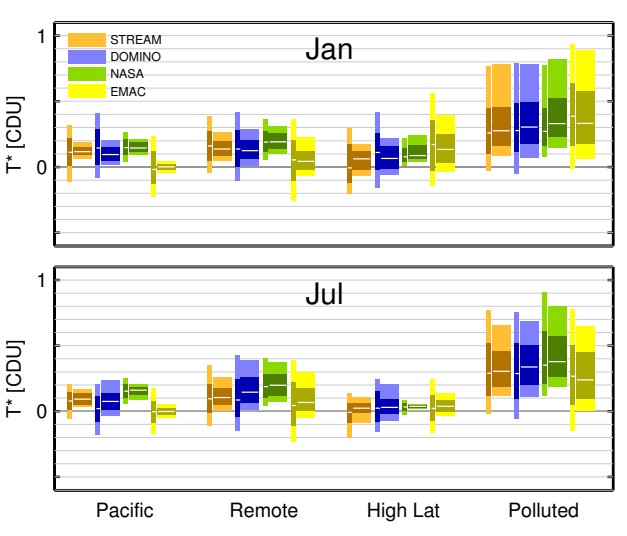

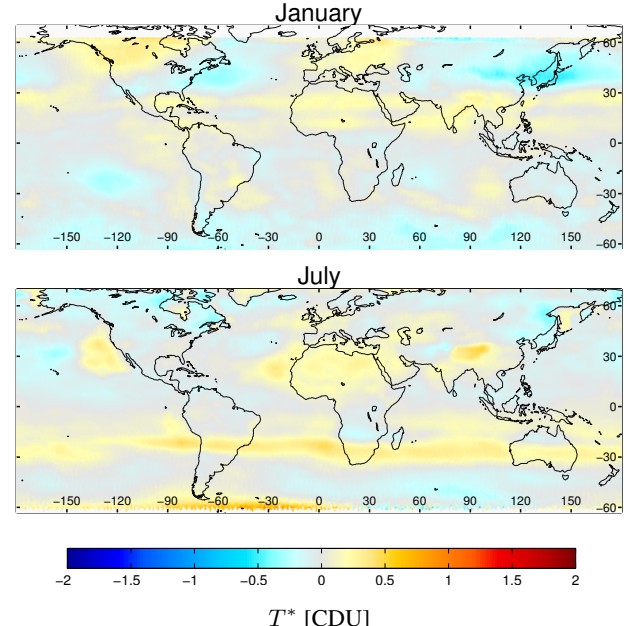

**Figure 8.** Regional statistics of OMI tropospheric residues $T^*$ from different STS algorithms for January (top) and July (bottom) 2005. Note that the values for STREAM slightly differ from Fig. 6, as here only coincident satellite pixels of STREAM, DOMINO and NASA are included.

**Figure 9.** Monthly mean difference of tropospheric residues $T^*$ from DOMINO and STREAM for OMI measurements in January (top) and July (bottom) 2005.



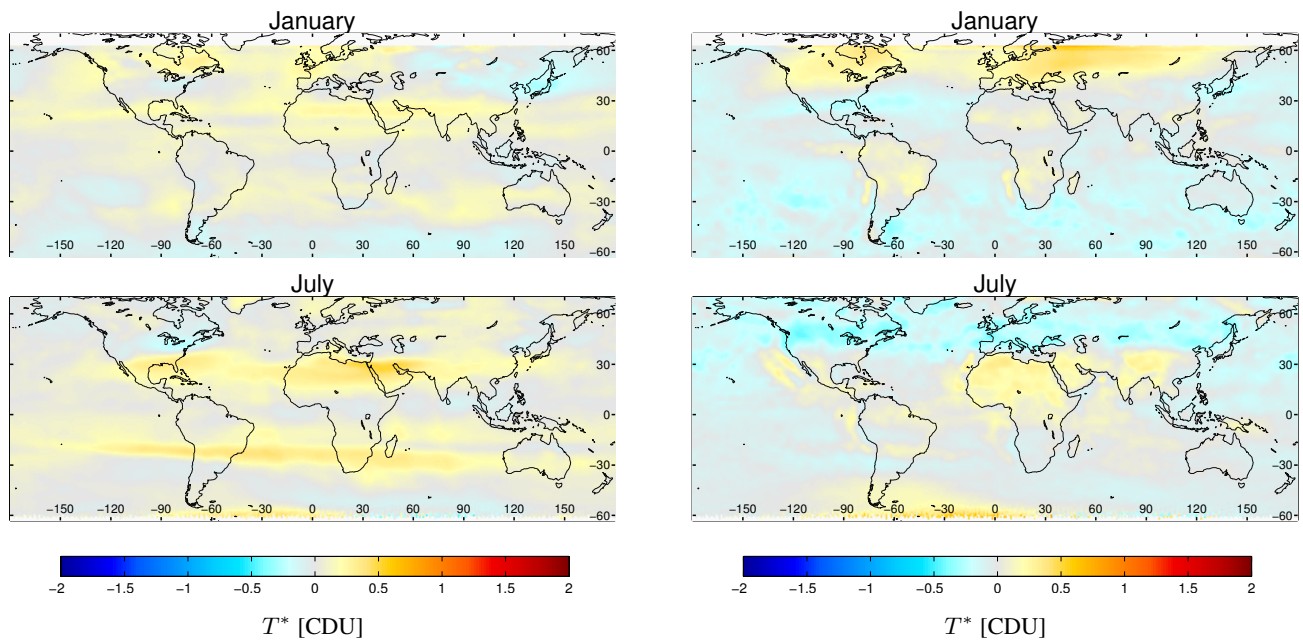

**Figure 10.** Monthly mean difference of tropospheric residues $T^*$ from NASA and STREAM for OMI measurements in January (top) and July (bottom) 2005.

**Figure 11.** Monthly mean difference of tropospheric residues $T^*$ from STS$_{EMAC}$ and STREAM for OMI measurements in January (top) and July (bottom) 2005.





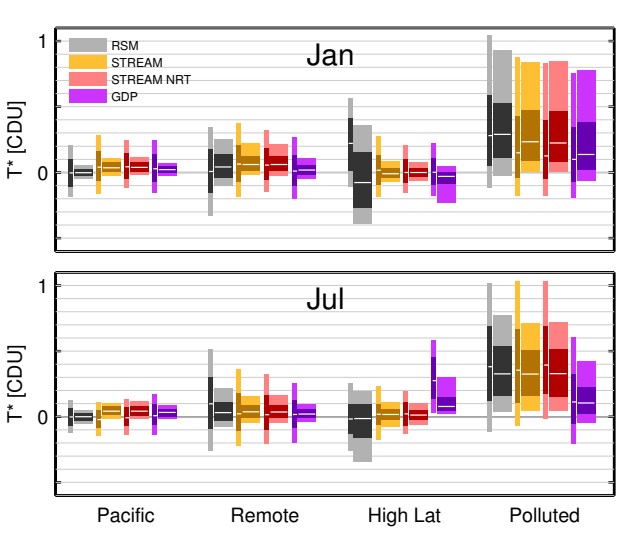

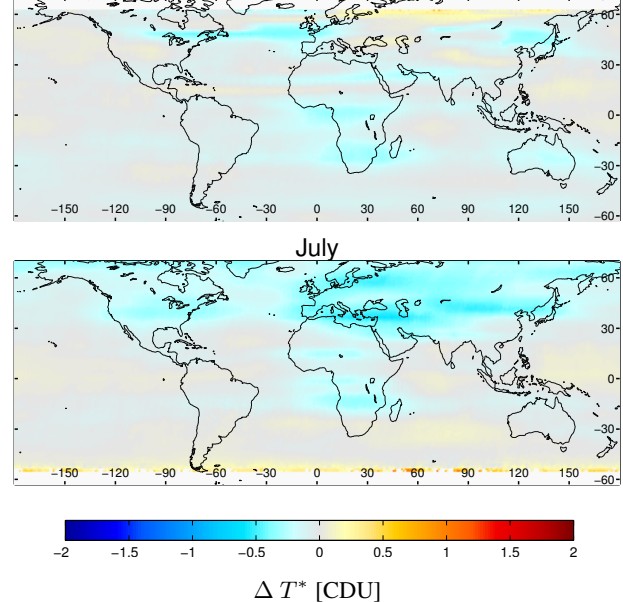

**Figure 12.** Regional statistics of GOME-2 tropospheric residues $T^*$ from different algorithms for January (top) and July (bottom) 2010. Conventions as in Fig. 6.

**Figure 13.** Monthly mean difference of tropospheric residues $T^*$ from GDP 4.7 and STREAM for GOME-2 measurements in January (top) and July (bottom) 2010.





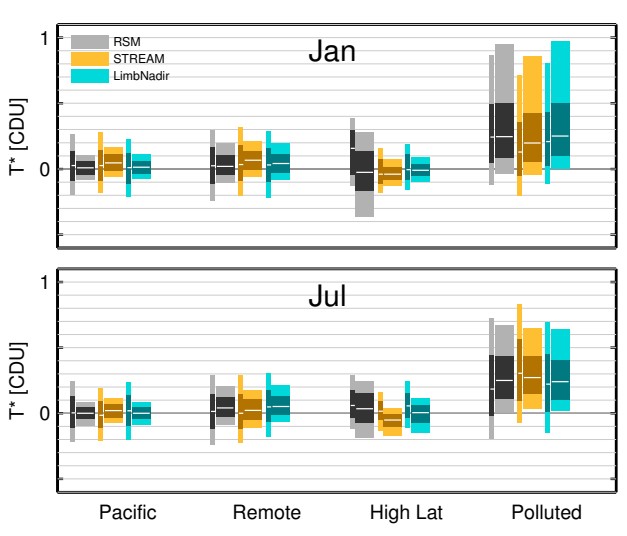

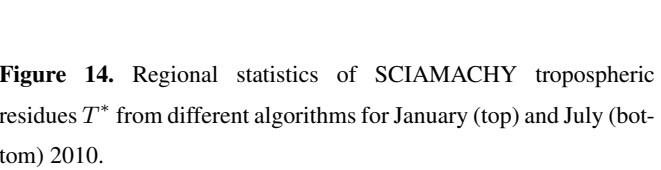

**Figure 14.** Regional statistics of SCIAMACHY tropospheric residues $T^*$ from different algorithms for January (top) and July (bottom) 2010.

Conventions as in Fig. 6.

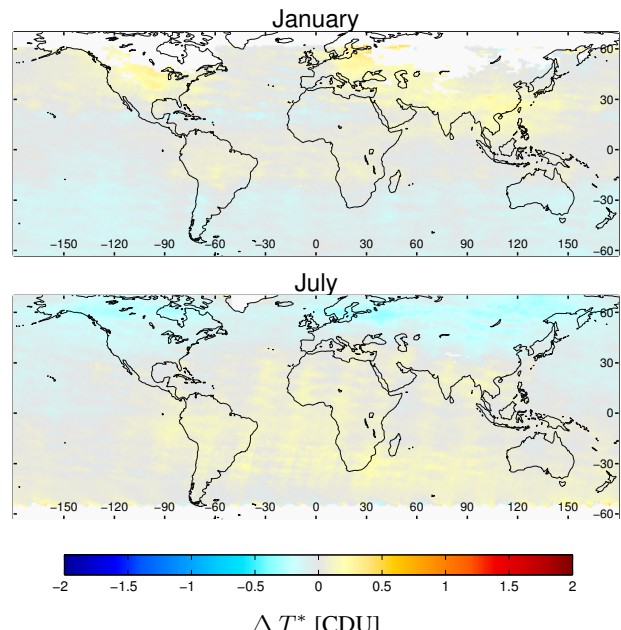

**Figure 15.** Monthly mean difference of tropospheric residues $T^*$ from LNM and STREAM for SCIAMACHY measurements in January (top) and July (bottom) 2010. Gaps at high latitudes in January are caused by respective gaps in the FRESCO cloud product.





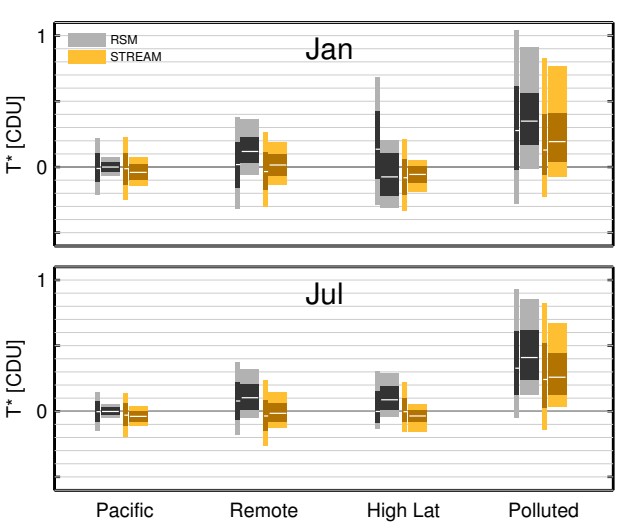

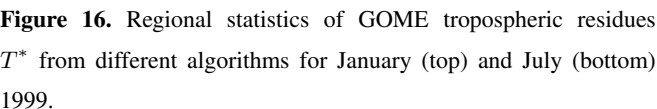

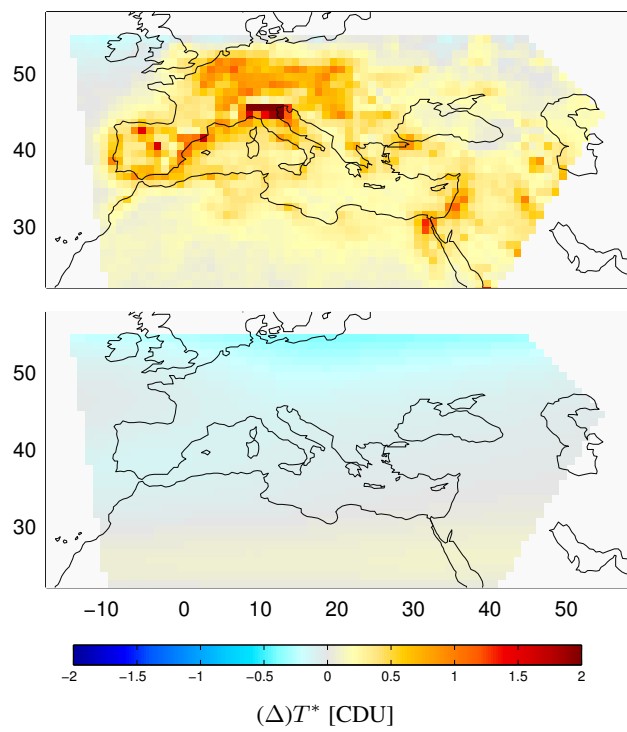

**Figure 16.** Regional statistics of GOME tropospheric residues
$T^*$ from different algorithms for January (top) and July (bottom)
1999.
Conventions as in Fig. 6.

**Figure 17.** Performance of STREAM on "S4 data" (i.e. OMI measurements clipped to the area covered by S4) for January 2005. The
top panel displays the resulting TR, the bottom panel shows the difference to the TR resulting from full OMI data as shown in Fig. 5.
The area covered by S4 in winter has been taken from Courrèges-Lacoste et al. (2010).



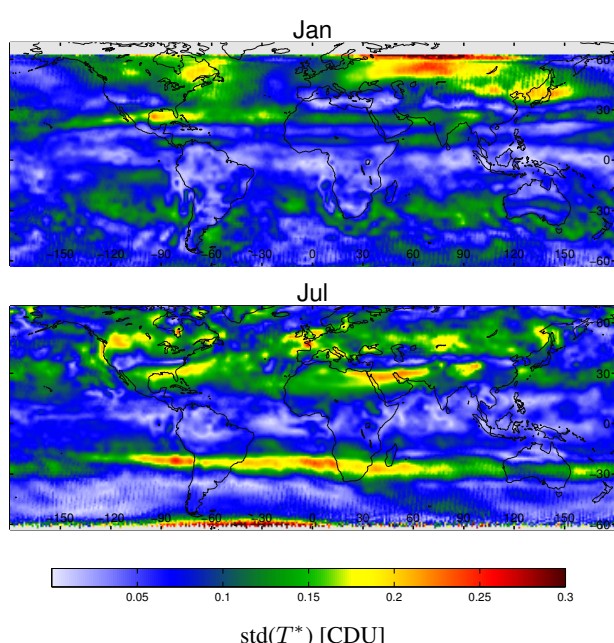

**Figure 18.** Standard deviation of monthly mean $T^*$ from different algorithms (STREAM, DOMINO, NASA, and $STS_{EMAC}$) for January (top) and July (bottom) 2005 (OMI).





**Table 1.** UV/vis satellite instruments compared or discussed in this study

| Acronym | Instrument | Launch | Footprint [km$^2$] | Earth coverage per day | Instrument reference | Data product used in this study | Data reference |
|---|---|---|---|---|---|---|---|
| GOME | Global Ozone Monitoring Experiment | 1995 | 40 × 320 | $^1/_3$ | Burrows et al. (1999) | TEMIS | Boersma and Eskes (2004) |
| SCIAMACHY | SCanning Imaging Absorption Spectro-Meter for Atmospheric CHartographY | 2002 | 30 × 60 | $^1/_6$ | Bovensmann et al. (1999) | MPI-C Mainz | Beirle and Wagner (2012) |
| GOME-2 | Global Ozone Monitoring Experiment-2 | 2006[a] | 40 × 80[b] | $^2/_3$ | Callies et al. (2000) | O3M SAF | Valks et al. (2015) |
| OMI | Ozone Monitoring Instrument | 2004 | 13 × 26[c] | 1[d] | Levelt et al. (2006) | NASA v3 / SP2 | Bucsela et al. (2013) |
| | | | | | | DOMINO v2 | Boersma et al. (2011) |
| TROPOMI | TROPOspheric Monitoring Instrument | 2016 | 7 × 7[c] | 1 | Veefkind et al. (2012) | | |
| Sentinel 4 | | 2021 | 7 × 7 | -[e] | Ingmann et al. (2012) | | |

[a] on Metop-A. A second GOME-2 instrument was launched 2012 on Metop-B, a third is planned to be launched on Metop-C in 2018.

[b] switched to 40 × 40 km$^2$ for GOME-2/Metop-A in Metop A+B tandem operation.

[c] at nadir.

[d] reduced coverage after row anomaly in 2007, see http://projects.knmi.nl/omi/research/product/rowanomaly-background.php.

[e] geostationary orbit: hourly coverage over Europe.





**Table 2.** Terms and abbreviations related to STREAM used in this study.

| Symbol | Abbrev. | Term | Description |
|---|---|---|---|
| $A$ | AMF | Air Mass Factor | Factor relating vertical to slant column density |
|  | CDU | Column Density Unit | Unit of column densities: $1 \times 10^{15}$ molecules cm$^{-2}$ |
| $G$ | CK | Convolution Kernel | Kernel used for weighted convolution; here: 2D Gaussian functions |
| $p_{\mathrm{cld}}$ | CP | Cloud pressure |  |
| $c$ | CRF | Cloud radiance fraction |  |
|  | LNM | Limb-Nadir-Matching | Stratospheric correction based on coincident stratospheric measurements in limb geometry (SCIAMACHY). |
|  | MRSM | Modified Reference Sector Method | A STS estimating stratospheric columns based on the total columns over "clean" regions, but allowing for longitudinal variations |
| $P$ |  | Pollution proxy | see section S2.2 in the supplement |
|  | RSM | Reference Sector Method | in general: a STS estimating stratospheric columns based on the total columns over a reference sector, assuming longitudinal homogeneity |
|  |  |  | in detail: $V_{\mathrm{strat}}$ is estimated over the Pacific (180°W to 140°W) |
| $S$ | SCD | Slant Column Density | concentration integrated along mean light path |
|  | STS | Stratosphere-troposphere separation | The procedure of separating the total column into stratospheric and tropospheric fractions |
| $T^{*}$ | TR | Tropospheric Residue | Difference of total and stratospheric column, Eq. 3 |
| $w$ |  | Weighting factor |  |
| $V_{\mathrm{trop}}$ | TVCD | Tropospheric Vertical Column Density | see Eq. 4 |
| $V$ | VCD | Vertical Column Density | Vertically integrated concentration |
| $V^{*}$ |  |  | Total VCD based on stratospheric AMF |
| $V_{\mathrm{strat}}$ |  |  | stratospheric VCD |
| $\vartheta$ | lat | latitude |  |
| $\varphi$ | lon | longitude |  |

*based on strat. AMF