# Peer review of "S1. Supplementary material"

_Atmospheric Measurement Techniques, 2015_

## Referee Comment (RC1) · Anonymous Referee #1 · 26 Feb 2016

This paper presents a good and thorough description of the STREAM algorithm for separating stratospheric and tropospheric NO2 in satellite retrievals. STREAM is a logical next step in the series of increasingly sophisticated RSM- and MRSM- type STS algorithms (e.g. Richter and Burrows, 2002 to Valks et al 2011 to Bucsela et al 2013). To estimate the stratosphere, the authors use measurement-based a priori pollution amounts, cloud conditions (both coverage and height) and then make an iterative correction for tropospheric contamination that may have been missed in the previous estimate. They do extensive comparisons of STREAM relative to other algorithms as applied to OMI, GOME-2 and SCIAMACHY as well as looking at results from an in-

dependent model estimate of stratospheric NO2 from the EMAC model. The paper is well written and generally well organized and worthy of publication after the authors address the major and minor concerns below.

Major comments:

1. The inherent question in any STS algorithm is whether features in the V* field belong in the stratosphere or troposphere. With its relatively smooth stratosphere (evident in Fig 3), STREAM correctly identifies some structures in unpolluted regions as tropospheric (e.g. tropospheric transport events). But it also folds some synoptic-scale stratospheric features into the troposphere. To clarify such ambiguities, it is essential to test an STS algorithm against truth by applying it to synthetic data. The authors perform such a test and show the effective retrieval errors. However given its importance, this section of the paper seems a bit brief. It would be useful to see how the weighting factors and convolution kernel sizes can be tuned to improve the retrieval. The section on parameter sensitivities indicates the magnitudes of the effects of parameter adjustments, but it could go further in determining which are actually better and why the authors believe STREAM's method is optimal. For example, could an increase in wcld (Fig. S9) help counter the low bias in T* seen in Fig S 18?

2. Stratospheric NO2 shows significant diurnal variation across an orbital swath. This is evident in the daily OMI DOMINO and NASA stratospheres when plotted orbit by orbit. From the description of STREAM, it is not apparent to me that the algorithm accounts for this, and therefore some of the stratospheric variation in each orbit will be aliased into the troposphere (this is also a problem in the RSM and in some of the other MRSM approaches). Please give an estimate of the magnitude of this effect and discuss the consequences of ignoring it, as well as the handling of orbital overlap – especially at mid- and high latitudes. A somewhat related question is how STREAM treats OMI pixels near the swath edges. Are these weighted differently?

3. In section 4.2 the authors describe a number of sensitivity studies. But sensitivity to

the pollution weights should also be included here since it seems likely these weights would have a large effect on the retrieval.

Additional comments:

1. (pp 5-6) I think the algorithm description needs a more detailed outline than the 2-step version given at the top of page 5. Please list each step explicitly, including the names of the three weighting factors, their product and the iteration involving the TR weights. This would improve clarity, since the order of presentation in the paper is not exactly the order of data processing.

2. (p 8) In section 2.2.3, setting-description #3 says that the minimum size of a region for considering a TR weight is a grid pixel and the immediately surrounding pixels. Is this also the maximum size?

3. (p 12) In section 3.1.1, the current version of the OMNO2 data product should be v2.1, not v3.

4. (p 12) Also in 3.1.1, the authors claim that new SCDs from van Geffen et al. (2015) and Marchenko et al. (2015) would not affect the TR. This is not true: the smaller SCDs will also decrease the retrieved tropospheric amounts.

5. (p 15) As described in section 4.1, Fig. 3 shows V* for two days along with the STREAM stratosphere. Please also show also the stratospheres for RSM, OMI DOMINO and OMI NASA on the same days, since these would help indicate how much of the V* structure is assigned to the stratosphere in each algorithm. It is difficult to determine this by looking only at the highly structured daily TR maps for the respective algorithms (see major comment 1)

6. (p 16) In the discussion of cloud weight (4.2.1, part (b)), why not use a larger wcld, which lowers Vstrat and increases the magnitude and standard deviation of T* – i.e. why does the algorithm have a lower wcld as its baseline?

7. (p 19) I suggest a wording change at the bottom of p19 (section 5.1.2): "...interprets

the difference between the full total column and the (small) tropospheric model as stratospheric column. . .”

8. (p 20) Regarding section 5.1.2, bullet point 2, the way the stratosphere is estimated need not affect NO2 retrievals by cloud slicing (see Choi et al., ACP 2014).

9. (p 20) In section 5.1.3, please state why an additive rather than multiplicative offset is applied to EMAC?

10. (p 22) Section 5.2, first sentence at the top of the page, a misspelling: “. . .extraordinarily high. . .”

11. (p 23) Section 5.3, minor wording changes: “. . .overall still works well. . .”, “. . .similar to GOME-2. . .”, “. . .similar to OMI or GOME-2. . .”

12. (p 26) Section 5.7 states that STS errors cannot be directly quantified since the “true” stratosphere is not known. As the authors have already shown, this can be known to some extent using synthetic data (major comment 1). Please include a comment here to this effect.

13. (p 26) Section 5.7, last sentence, a misspelling: “. . .focusing. . .”

14. (p 27) Section 6, Conclusions, 2nd paragraph, a misspelling: “. . .with a high weighting factor.”
* * *

---

## Editor Comment (EC1) · B. Veihelmann (Editor) · 11 Apr 2016

Main comments:

1. One of the strengths of STREAM is its capability of exploiting measurement data also over continents, for estimating Vstrat. The largest benefit with respect to other STS methods would hence be at clear sky continental scenes. Please clarify if that is correct. If so, a dedicated analysis of such scenes would be valuable, eg based on the synthetic data used.

2. STREAM differs from other STS methods in the sense that scenes with tropospheric

NO2 are not excluded but that a low weight is assigned. Please clarify whether this introduces a bias in Vstrat, and - if so - whether the tropospheric-residual-based weight mitigates such a bias. STREAM is tested in a scenario where only the pollution weights and the cloud weights are applied (Section 4.2.5), which would be the ideal test scenario to answer this question.

3. The pollution weight is derived from multi-annual mean tropospheric NO2 column data from SCIAMACHY (Beirle and Wagner, 2012). Please clarify how spatially smooth these data are, in the context of the width of the convolution kernels, and vis-à-vis the application of STREAM to higher spatially resolving instrumentations.

Minor comments:

- The assumptions made for the stratospheric Air Mass Factor should be specified (it is expected to be approximated by the geometric AMF).

- p4, line 16: Typo: complImentary → complEmentary
* * *

---

## Referee Comment (RC2) · Anonymous Referee #2 · 25 Apr 2016

The manuscript "The STRatospheric Estimation Algorithm from Mainz (STREAM): Estimating stratospheric NO 2 from nadir-viewing satellites by weighted convolution" by Beirle et al. is a very thorough description of a new algorithm for the separation of stratosphere and troposphere in the space-borne measurements of tropospheric NO$_2$. It is very well written and a pleasure to read. I recommend the manuscript to be published in *Atmospheric Measurement Techniques*. However, in order to further improve the manuscript, I suggest addressing the following minor comments in a revised version of the manuscript:

[Figure]

- The number 1E14 molec/cm$^2$ is mentioned several times as tropospheric background NO$_2$ columns over the remote Pacific Ocean, and cite a publication by Valks et al for reference. However, it should be noted that other studies (Martin et al., doi:10.1029/2001JD001027, Fig. 8; Hilboll et al., 2013, doi:10.5194/amt-6-565-2013, Fig. 5) derive significantly higher background values over the Pacific. It would be good if the authors could acknowledge that the number they use is at the lower end of a range of values proposed by previous studies.

- p. 7, l. 4-5: The mean spatial distribution does not reflect the pollution probability, as claimed by the authors. E.g., the same mean value can be caused by a single extreme pollution event in an otherwise clean region, or by moderate, constant in time, pollution levels. So the notion of *probability* should not be used in this context.

- p. 7, l. 5-6: The authors should specify if the *multi-annual mean trop. NO$_2$ column* the use does include the seasonal cycle, i.e., if they have one "multi-annual mean" per month. If not, the authors should clarify how the seasonal cycle is being considered in the weight calculation.

- p. 7, l. 10: It would help the reader if the authors could give a range for the pollution proxy $P$. Otherwise, it is impossible to grasp how large $w_{pol}$ is in comparison to the other weights.

- p. 8, l. 12: The authors should explain why measurements where the strat. contribution has been overestimated should *contribute more strongly to the strat. estimate*.

- p. 9, l. 2: The reference to "S4.2.3" is wrong.

- p. 9, l. 3-6: The example of pixels over U.S., Europe, central Africa, and China leading to low $w_{TR}$ is not helping, since without further information, the reader

has to assume that these regions already have low weight due to $w_{pol}$. As the differentiation between the pollution and the trop. res. weights is not immediately clear to the reader anyways, it might be a good idea to find an example of unusually high polluted regions, which would not have been assigned low weights by using $w_{pol}$ alone.

- p. 10, l. 1: If the authors set $W_{ij} = 0$ in case of measurement gaps, then $V_{ij}$ as defined by the authors is not defined. Shouldn't it be enough to set $C_{ij} = 0$? Otherwise, please amend Eq. 11 so that it yields a well-defined $V_{ij}$ everywhere.

- p. 10, l. 4: The authors should clarify if the 2D Gaussian they use as CK is defined in degree-space or in kilometer-space. If it is defined in degree-space, they should justify the resulting inconsistency depending on latitude.

- p. 13, l. 31: There's a spurious "see" in the refernce to Jöckel et al.

- p. 14, l. 6: The authors should specify how the EMAC model determines the tropopause height, i.e., thermal, dynamical, . . . criterion?

- p. 17, l. 5: Currently, the manuscript states that "the final Vstrat [. . . ] as weighted mean of both [CKs]". This is not really precise, it should rather say that the final Vstrat is the weighted mean of Vstrat calculated with both CKs.

- p. 18, l. 6: The authors write of "the" small-scale structures of strat. $NO_2$ in EMAC. It would be good to elaborate a bit on the "the", i.e., in which regions, in which months, . . . I suspect that these structures are mostly there at low latitudes, but it would be good if the authors could be more explicit about this.

- p. 18, l. 10-11: The authors should clarify if the "remaining biases" are low or high biases.

- p. 18, l. 26: "meaningful" has an extra "l"

- p. 19, l. 11-12: The authors should comment on whether they expect higher variability of $T^*$ in DOMINO or STREAM.

- p. 19, l. 17: underestimation "by", not "of" DOMINO

- p. 21, l. 9-11: Might the longitudinal dependency in STS$_{EMAC}$ partly be caused by the temporal sampling of the EMAC model? The LT difference between East and West end of the OMI swatch is rather large, and using a fixed EMAC time might introduce longitudinally varying biases

- p. 21, l. 28-29: GOME/SCIAMACHY/GOME-2 might have systematic differences in the CTP products, which might in turn influence the STS results. Furthermore, the spatial resolution of the different instruments should lead to different pdfs for the cloud fraction, again possibly influencing the STS results. It would be good if the authors could comment on this issue.

- p. 22, l. 24: "MOZART" should be replaced with "MOZART-2"

- p. 24, l. 20: Also, the smaller pixel size will lead to a higher fraction of purely clouded pixels. The authors should briefly comment on the implications.

- Sect. 5.5.2: Sentinel-4 is the name of the satellite, not the instrument. The authors should instead write something like "UVN onboard Sentinel-4" or the like.

- p. 26, l. 8: Please add "on average" before "negative $T^*$, because single negative $T^*$ are expected, as the authors already noticed elsewhere.

- p. 26, l. 31-32: The authors should note that applying STREAM to other trace gases, where the bulk of the profile cannot be expected to lie within the boundary layer but rather in the same altitude ranges as clouds is at least challenging.

[Figure]

- p. 27, l. 21-24: The authors should think about ways in which the LT of measurement being in the morning could hinder STREAM for GOME/SCIAMACHY/GOME-2.

- Fig. 8: I cannot understand how the 90% percentile $T^*$ over polluted regions can still be lower than 1 CDU. Is this really correct? Furthermore, I have trouble deriving the minimum in the $T^*$ difference over China/Japan in Fig. 9 from the statistics given in this Fig. 8.

  After seeing the definition of "polluted" in Fig. S7, this becomes clear; but it points to the misleading label "polluted" in this context.

- Figs 9, 10, 11, S19, S20, S23, S24, S26, S27b, S30-S31, S33 should have $\Delta T^*$ as colorbar label instead of $T^*$.

- Fig. S9: Isn't increasing the cloud weight by a factor of 10 equivalent to increasing the total weight by a factor of 10, due to the definition of the total weight in Eq. 8? Why should this Figure then say something about the cloud weight in particular?

- The authors should introduce the OMI instrument before making reference to it from p. 8 onwards. In the current manuscript, OMI is only introduced later, in Sect. 3.1.

---

## Author Comment (AC1) · 20 May 2016

We would like to thank the reviewer for his/her constructive comments and suggestions. Below we reply to the raised issues point by point.
Figure numbers refer to the discussion version of manuscript and supplement.

Major comments:
1. The inherent question in any STS algorithm is whether features in the V* field belong in the stratosphere or troposphere. With its relatively smooth stratosphere (evident in Fig 3), STREAM correctly identifies some structures in unpolluted regions as tropospheric (e.g. tropospheric transport events). But it also folds some synoptic-scale stratospheric features into the troposphere. To clarify such ambiguities, it is essential to test an STS algorithm against truth by applying it to synthetic data. The authors perform such a test and show the effective retrieval errors. However given its importance, this section of the paper seems a bit brief. It would be useful to see how the weighting factors and convolution kernel sizes can be tuned to improve the retrieval. The section on parameter sensitivities indicates the magnitudes of the effects of parameter adjustments, but it could go further in determining which are actually better and why the authors believe STREAM's method is optimal. For example, could an increase in wcld (Fig. S9) help counter the low bias in T* seen in Fig S 18?

Reply: The correct identification and removal of synoptic scale stratospheric features is indeed quite challenging for all algorithms based on modified RSM. We have performed the stratospheric estimate on synthetic data in order to quantify this effect and other possible shortcomings, like the tropospheric background, and generally the accuracy of STREAM. We agree that the idea of optimizing the weighting factors for the synthetic data is quite tempting. As stated by the reviewer, such an approach would indeed favor a larger $w_{cld}$, which would also increase the tropospheric background in the Pacific for OMI (see Fig. S11). However, we refrain from this approach, as the results for SCIAMACHY and GOME-1/2 would become worse due to the different dependencies of total columns on cloud fraction. The chosen definitions for weighting factors therefore remain somewhat arbitrary. However, as demonstrated, the impact of changes of these definitions on resulting TR is small.
In the revised manuscript, we have extended section 4.3 according to detailed comments of reviewer 2. In addition, we conclude the section with a new paragraph:
*"The application of STREAM to synthetic data thus provides a valuable estimate of the algorithm's accuracy. One might think of using the synthetic data for optimizing the definition of weighting factors being the next step forward. However, we refrain from doing so due to some contradictory results for different instruments. Concretely, the remaining bias in TR for synthetic data of about 0.1 CDU could be further reduced by increasing $w_{cld}$. This, however, has adverse effects on SCIAMACHY and GOME results (see sections 5.3 and 5.4)."*

2. Stratospheric NO2 shows significant diurnal variation across an orbital swath. This is evident in the daily OMI DOMINO and NASA stratospheres when plotted orbit by orbit. From the description of STREAM, it is not apparent to me that the algorithm accounts for this, and therefore some of the stratospheric variation in each orbit will be aliased into the troposphere (this is also a problem in the RSM and in some of the other MRSM approaches). Please give an estimate of the magnitude of this effect and discuss the consequences of ignoring it, as well as the handling of orbital overlap – especially at mid- and high latitudes. A somewhat related question is how STREAM treats OMI pixels near the swath edges. Are these weighted differently?

Reply: As stated in section 2.3, each satellite pixel is assigned to one grid pixel (i,j) according to its center coordinates.

The satellite pixels are all treated equally, independent of viewing angle or pixel size. However, since the center of OMI's swath contains many more satellite pixels per grid pixel than the swath edge, the smaller center pixels dominate the weighted convolution procedure. In case of orbital overlap, as for OMI, a grid pixel might contain satellite pixels with different local time. This effect is neglected in STREAM. According to the reviewers request, we have estimated the magnitude of this effect by calculating the mean total NO2 VCD separately for the different OMI viewing zenith angles (directly related to local time). We refer to this aspect in section 2.4 (Data processing) of the revised manuscript, and have added this investigation to section S 2.4 of the Supplement:

Section 2.4:
*"STREAM estimates stratospheric fields and tropospheric residues for individual orbits, using $NO_2$ measurements of the dayside of the orbit. Note that the effect of changes of local time on stratospheric $NO_2$ across orbit is generally low (see section S2.4 in the Supplement) and is thus neglected within STREAM."*

*"S 2.4 Data Processing*
*Diurnal variation of stratospheric $NO_2$*

[Figure]

*Fig. A: Dependency of OMI total VCD V\* on cross-track position (compared to nadir) at different latitudes in January 2005.*

*In case of orbital overlap, as for OMI, a grid pixel might contain satellite pixels with different local time. This effect is neglected in STREAM. We have estimated the magnitude of this effect by calculating the mean total $NO_2$ VCD separately for the different OMI viewing zenith angles, which are directly related to local time (LT) (see Fig. A).*
*For low latitudes, the effect is negligible. Only for high latitudes (>50°), the effect can exceed ±0.2 CDU at the swath edges. Consequently, if individual orbits are considered, a small cross-track dependency of TR could be observed for high latitudes, which is actually caused by the LT dependency of stratospheric $NO_2$. For gridded data, however, where OMI orbits significantly overlap, the effect (and its impact on STREAM performance) is generally negligible."*

3. In section 4.2 the authors describe a number of sensitivity studies. But sensitivity to the pollution weights should also be included here since it seems likely these weights would have a large effect on the retrieval.
Reply: We have performed a sensitivity study on the impact of $w_{pol}$:

[Figure]

*Fig. B: Regional statistics of OMI TR from STREAM for different settings for $w_{pol}$ for January (top) and July (bottom) 2005.*

The figure is included in the revised supplement (section S4.2.6).
In the manuscript, we have added

*"4.2.6 Impact of pollution weight*
*The impact of pollution weight is investigated by multiplying $w_{pol}$ (where different from 1, compare Fig. 2(a)) by 0.1 ("low $w_{pol}$") or 10 ("high $w_{pol}$"). In the first case, the resulting pollution weight over most continents is below 0.01, while in the second case, it is increased to 1 (meaning that $w_{pol}$ is switched off) except for industrialized pollution hotspots.*
*In remote regions, the change of $w_{pol}$ has almost no impact. In potentially polluted regions, the impact is only moderate as well. Low $w_{pol}$ does not differ much from the baseline, as the latter already assigns rather low weighting factors to potentially polluted pixels; a further decrease by factor 0.1 thus does not change much.*
*Only for high $w_{pol}$, a significant change of TR can be seen; in this case, the inclusion of more partly polluted observations causes a high bias in the stratospheric estimate and the resulting TR are biased low by almost 0.1 CDU in winter."*

Additional comments:
1. (pp 5-6) I think the algorithm description needs a more detailed outline than the 2-step version given at the top of page 5. Please list each step explicitly, including the names of the three weighting factors, their product and the iteration involving the TR weights. This would improve clarity, since the order of presentation in the paper is not exactly the order of data processing.
Reply: We have slightly extended the algorithm description in section 2:
*"STREAM consists basically of two steps:*
*1.  A set of weighting factors  is calculated for each satellite pixel: a "pollution weight" that reduces the contribution of potentially polluted pixels, a "cloud weight" that increases the contribution of cloudy observations, and a "tropospheric residue weight" that adjusts the total weight in case of exceptionally large or negative tropospheric residues. The product of these weighting factors determines to what extent the associated NO2 total columns contribute to the estimated stratospheric field (Sect. 2.2). *
*2. Global maps of stratospheric NO2 are determined by applying weighted convolution (Sect. 2.3)."*
This introductory paragraph is meant as high-level overview and thus does not provide more details, but refers to the respective subsections.

2. (p 8) In section 2.2.3, setting-description #3 says that the minimum size of a region for considering a TR weight is a grid pixel and the immediately surrounding pixels. Is

this also the maximum size?

Reply: Within the TR iteration, the criteria 1-3 are checked for each grid pixel ij. I.e., the size of the region that will be modified by $w_{TR}$ is not restricted and can be quite large, as long as the 2nd criterion ($|T*|>0.5$ CDU) is fulfilled as well.

We have modified the formulation of criterium 3 in the revised manuscript to make the motivation for this additional condition more clear:

"*3. $w_{TR}$ is only applied for grid pixels where the adjacent grid pixels exceed the threshold as well. By this additional condition it is guaranteed that a single outlier in the satellite measurements cannot trigger $w_{TR}$, as any satellite measurement is assigned to one grid pixel (see section 2.3).*"

3. (p 12) In section 3.1.1, the current version of the OMNO2 data product should be v2.1, not v3.

Reply: The data files we have downloaded from http://mirador.gsfc.nasa.gov/ all include "v003" in the filename, and on http://mirador.gsfc.nasa.gov/collections/OMNO2__003.shtml, it is explicitly stated that the dataset version is 003. We have thus modified "*v3*" to "*v003*".

4. (p 12) Also in 3.1.1, the authors claim that new SCDs from van Geffen et al. (2015) and Marchenko et al. (2015) would not affect the TR. This is not true: the smaller SCDs will also decrease the retrieved tropospheric amounts.

Reply: Note that we did not claim that the updated SCDs will not affect the TR, but the performance of STREAM.

We have modified the section to

"*However, such an overall bias will be interpreted as stratospheric feature by STREAM and thus does not affect its  performance . (the same holds for the operational NASA and TEMIS STS algorithms). Still, the resulting TRs are expected to decrease slightly as the bias decreases for larger SCDs (see Marchenko et al. (2015), Figure 3 therein).*"

5. (p 15) As described in section 4.1, Fig. 3 shows V* for two days along with the STREAM stratosphere. Please also show also the stratospheres for RSM, OMI DOMINO and OMI NASA on the same days, since these would help indicate how much of the V* structure is assigned to the stratosphere in each algorithm. It is difficult to determine this by looking only at the highly structured daily TR maps for the respective algorithms (see major comment 1)

Reply: Following the reviewer's suggestion, we have added maps of stratospheric NO2 from RSM, OMI NASA, OMI DOMINO, and EMAC to Figure 3 for the two selected days.

[Figure]

*Fig. 3: Total OMI VCD V\* (top) and the resulting stratospheric estimate $V_{strat}$ from RSM and STREAM for 1 January (left) and 1 July (right) 2005. Stratospheric estimates from other algorithms are included as well for comparison (see section 5.1).*

6. (p 16) In the discussion of cloud weight (4.2.1, part (b)), why not use a larger wcld, which lowers Vstrat and increases the magnitude and standard deviation of T\* – i.e. why does the algorithm have a lower wcld as its baseline?

Reply: As explained in the reply to the major comment 1, a higher $w_{cld}$ would indeed be preferable for OMI. However, this would cause artefacts for SCIAMACHY and particularly for GOME due to the different dependencies of total columns on cloud properties.

We have added a discussion on this to the end of section 4.2.1:

*"Following the argument that cloudy observations provide a direct measurement of the stratospheric column, a higher cloud weight would be expected to be more favorable, and to result in higher tropospheric background over the Pacific. This is indeed observed for OMI. For other satellite instruments, however, results are somewhat contradictory (see sections 5.3, 5.4 and 6). Thus, the definition of $w_{cld}$ in Eq.6 is kept as compromise in order to have common algorithm settings across different satellite platforms."*

7. (p 19) I suggest a wording change at the bottom of p19 (section 5.1.2): "…interprets the difference between the full total column and the (small) tropospheric model as stratospheric column…"

Reply: We have seized the reviewer's suggestion and slightly changed the sentence to
*"...interprets the difference between the total column and the (small) modelled tropospheric column as stratospheric column..."*

Reply: Choi et al. (2014) do not use the operational tropospheric product, but instead calculate dedicated clouded AMFs and derive a stratospheric estimate by themselves.
But if any user would investigate the dependency of the official tropospheric NO2 column on cloud height, results would probably be quite different.

Reply: We have applied an additive offset correction as simplest procedure. If instead a multiplicative adjustment is performed, results are hardly affected. We note this in the revised manuscript.

Corrected.

Reply: We have revised the manuscript accordingly.

Reply: We agree and have extended the section to
*"The uncertainty of STS can often not be directly quantified, as the "true" stratospheric 4D concentration fields are not known. One approach to assess the STS performance is the usage of synthetic data, as in section 4.3.*
*In addition, the TR can be used to evaluate the plausibility of the stratospheric estimate and to derive realistic uncertainties:"*
In addition, we refer to the results of the synthetic study when discussing standard deviations and typical differences between different algorithms:
*"Overall, the standard deviation of TR from different STS is low (typically <0.1 CDU, and below <0.2 CDU for most parts of the world). It is thus consistent with the uncertainty estimates of stratospheric columns given in literature (Boersma et al. (2011): 0.15-0.25 CDU (SCD); Valks et al. (2011): 0.15-0.3 CDU (VCD); Bucsela et al. (2013): 0.2 CDU (VCD)) and with the magnitude of systematic deviations found in the study on synthetic data (section 4.3)"*

Reply: Corrected.

Reply: Corrected.

---

## Author Comment (AC2) · 20 May 2016

We would like to thank the editor for his constructive comments and suggestions. Below we reply to the raised issues point by point. Figure numbers refer to the discussion version of manuscript and supplement.

Main comments:
1. One of the strengths of STREAM is its capability of exploiting measurement data also over continents, for estimating Vstrat. The largest benefit with respect to other STS methods would hence be at clear sky continental scenes. Please clarify if that is correct. If so, a dedicated analysis of such scenes would be valuable, eg based on the synthetic data used.

Reply: One of STREAM's features is the inclusion of clouded pixels (via $w_{cld}$), which provide additional supporting points over (not too polluted) continental regions, and thus avoids interpolation errors or wave fitting artefacts. This is expected to result in more realistic stratospheric estimates over continents and thus more accurate tropospheric residues, not only for clear sky.

In section 5.2.2, STREAM is compared to the GOME-2 GDP 4.7, which is based on a modified RSM with a pollution mask. Figure 12 clearly shows that the GDP 4.7 TRs are biased low over continents, and that STREAM overcomes this bias.

In addition, sensitivity study 4.2.1 clearly demonstrates the improvement due to the application of $w_{cld}$; if the latter is omitted, TR over potentially polluted regions (which are at large part congruent with continents) is obviously biased low (see Fig. S11 of the supplement).

2. STREAM differs from other STS methods in the sense that scenes with tropospheric NO2 are not excluded but that a low weight is assigned. Please clarify whether this introduces a bias in Vstrat, and - if so - whether the tropospheric-residual-based weight mitigates such a bias. STREAM is tested in a scenario where only the pollution weights and the cloud weights are applied (Section 4.2.5), which would be the ideal test scenario to answer this question.

Reply: Generally, all modified RSM methods estimate the stratospheric column from total column measurements. Thus, the estimated stratospheric field is generally expected to be biased high due to the tropospheric background (if the latter is not explicitly corrected for based on CTMs or other a-priori). Consequently, TR are expected to be biased low. The effect is dampened in STREAM by the high weight of clouded pixels. But still, TR over the Pacific and remote regions is about 0.1 CDU for OMI, and less for GOME 1/2 and SCIAMACHY, while models expect a slightly higher background (see the reply to the first remark of reviewer 2).

This topic is mentioned in the introduction (item (a) on 2$^{nd}$ page), but still we have tried to further clarify this fundamental aspect in the revised manuscript by extending the end of the introduction:

*"In particular, clouded observations are weighted high, as they provide direct measurements of the stratospheric field. This approach dampens the small but systematic high bias of stratospheric columns estimated from total column measurements and the resulting low bias of tropospheric columns."*

However, the TR weight cannot correct this small systematic bias, as it is only applied if the mean TR has an absolute value > 0.5 CDU (otherwise, noise would introduce severe artefacts via $w_{TR}$, see section 2.2.3). We have modified the end of section 2.2.3 accordingly:

*"Note that due to the threshold of 0.5 CDU (criterion 2), $w_{TR}$ cannot correct small biases such as the expected low bias in TR caused by estimating the stratospheric column from total column measurements."*

3. The pollution weight is derived from multi-annual mean tropospheric NO2 column data from SCIAMACHY (Beirle and Wagner, 2012). Please clarify how spatially smooth these data are, in the context of the width of the convolution kernels, and vis-à-vis the application of STREAM to higher spatially resolving instrumentations.

Reply: The definition of the pollution proxy P is provided in section S2.21 of the supplement and displayed in Figure S2. We have clarified this in the revised manuscript:

*"Details on the definition of P are given in the supplement (sect. S2.2.1), and P is displayed in Fig. S2d."*

As P is intended to indicate *potentially* polluted regions, and the original SCIAMACHY climatology is intentionally smeared out by convolution, SCIAMACHY's spatial resolution is not critical.

We have extended section S2.2.1 accordingly:

*"1. Grid pixels with a mean TVCD below 1 CDU are removed (Fig. S2b).*

*2. The resulting clipped climatology is smoothed by convolution with a 2D-Gaussian with $\sigma = 2°$ (Fig. S2c).*

*3. For the pollution proxy P, values between 0 and 1 CDU are set to 1 CDU. By this operation, a "safety margin" of potentially polluted areas is created (Fig. S2d).*

*Note that due to steps 2&3, the initial spatial resolution of SCIAMACHY is fully sufficient for the definition of P (and thus $w_{pol}$) even for applications of STREAM to instruments with better spatially resolution."*

Minor comments:
- The assumptions made for the stratospheric Air Mass Factor should be specified (it is expected to be approximated by the geometric AMF).

Reply: Stratospheric AMFs have been taken from the same data source as the NO2 columns, as listed in Table 1.

We have extended section 3.1 accordingly:

*"Table 1 summarizes the characteristics of the instruments  and provides references to the data products used in this study, from which the total SCD, the stratospheric AMF, and the cloud fraction/cloud top height are taken as input for STREAM. "*

- p4, line 16: Typo: complImentary → complEmentary

Reply: Done.

---

## Author Comment (AC3) · 20 May 2016

We would like to thank the reviewer for his/her constructive comments and suggestions. Below we reply to the raised issues point by point.
Figure numbers refer to the discussion version of manuscript and supplement.

• The number 1E14 molec/cm2is mentioned several times as tropospheric background NO2 columns over the remote Pacific Ocean, and cite a publication by Valks et al for reference.
However, it should be noted that other studies (Martin et al., doi:10.1029/2001JD001027, Fig. 8; Hilboll et al., 2013, doi:10.5194/amt-6-565-2013, Fig. 5) derive significantly higher background values over the Pacific. It would be good if the authors could acknowledge that the number they use is at the lower end of a range of values proposed by previous studies.

Reply: We agree that the discussion of the tropospheric background was too simplified in the original manuscript. Thus we have revised the manuscript accordingly:

a) in the introduction, we have modified line 22 to

*"Neglecting the tropospheric background results in tropospheric columns that are biased low by* some $10^{14}$ *molec cm*$^{-2}$ *(Martin et al., 2002; Valks et al., 2011; Hilboll et al., 2013)"*

b) in section 5.6, we have extended the discussion respectively:

*"As (M)RSMs usually estimate the stratospheric column based on total column measurements over clean regions, they generally miss the (small) tropospheric background of the order of* some *0.1 CDU. Several (M)RSMs explicitly correct for this effect based on a-priori tropospheric background columns (Martin et al., 2002; Valks et al., 2011; Bucsela et al., 2013). In case of STREAM, however, cloudy pixels, which allow a direct measurement of the actual stratospheric column (except for the small tropospheric column above the cloud), are emphasized. Thus, an additional tropospheric background correction*  *should be unnecessary.*

*Accordingly, in case of OMI, TR from STREAM are about 0.1 CDU over clean regions, similar as for TR from DOMINO and NASA. This is close to the a-priori value chosen by Valks et al. (2011), but below the values given in Martin et al. (2002) (about 0.15-0.3 CDU, assuming a tropospheric AMF of 2) and Hilboll et al. (2013) (0.1 up to >0.6 CDU; note, however, that the high values are only reported at higher latitudes in winter, when the ratio $AMF_{strat}/AMF_{trop}$ is almost 2 (Fig. S1); thus the large discrepancy is at least partly resolved if the TR is transferred in a TVCD via equation 4.)"*

• p. 7, l. 4-5: The mean spatial distribution does not reflect the pollution probability, as claimed by the authors. E.g., the same mean value can be caused by a single extreme pollution event in an otherwise clean region, or by moderate, constant in time, pollution levels. So the notion of probability should not be used in this context.

Reply: We agree that the term "probability" is misleading here.
We have revised the respective sentence to

*"We thus define a pollution weight $w_{pol}$ based on our a-priori knowledge about the mean spatial distribution of tropospheric NO$_2$, reflecting*  *potentially polluted regions."*

• p. 7, l. 5-6: The authors should specify if the multi-annual mean trop. NO2 column the use does include the seasonal cycle, i.e., if they have one "multi-annual mean" per month. If not, the authors should clarify how the seasonal cycle is being considered in the weight calculation.

Reply: The basis of the calculation of P is the mean NO$_2$ column density from SCIAMACHY. So far, no seasonality is accounted for.

This might be modified in a future implementation.

However, given the weak dependency of the results on the definition of $w_{pol}$ (see reply to major comment #3 of reviewer #1), the impact of a seasonally varying pollution weight is expected to be negligible.

We have added this discussion to section S2.2.1 of the Supplement:

*"So far, no seasonality of $NO_2$ is considered in the definition of P. This could be added to a future version, but the impact on STREAM is expected to be low (compare section 4.2.6)."*

• p. 7, l. 10: It would help the reader if the authors could give a range for the pollution proxy P. Otherwise, it is impossible to grasp how large wpol is in comparison to the other weights.

Reply: The pollution proxy P is displayed in Fig. S2d. Values are about 1 CDU for most remote continental regions up to >3 CDU for central Europe and the US Eastcoast and >6 for China. According to Eq. 5, the resulting pollution weights for these numbers are 0.1, 0.0037, and 0.0005 (compare Fig. 1a).

In the revised manuscript, we have added

*"Details on the definition of P are given in the supplement (sect. S2.2.1), and P is displayed in Fig. S2d."*

to the end of page 7, line 9.

• p. 8, l. 12: The authors should explain why measurements where the strat. contribution has been overestimated should contribute more strongly to the strat. estimate.

Reply: In cases of negative TR, the estimated stratospheric column is larger than the total column measurement itself. This overestimation of the stratospheric column is caused by neighboring observations via weighted convolution. By increasing the weight of the local measurement, its impact on the stratospheric estimate is enlarged, resulting in a lower stratospheric estimate, and a higher (ideally>0) TR.

We have clarified this in the revised manuscript:

*"As negative columns are nonphysical, $T^* < 0$ indicates that the stratospheric field has been overestimated. This happens if the weighted convolution with neighboring pixels with high total columns causes the estimated stratosphere to be even higher than the local total columns. In order to avoid this effect, the respective local total columns should be assigned a higher weighting factor such that they contribute more strongly to the stratospheric estimate."*

• p. 9, l. 2: The reference to "S4.2.3" is wrong.

Reply: We have corrected this to "*S4.2.5*".

• p. 9, l. 3-6: The example of pixels over U.S., Europe, central Africa, and China leading to low wTR is not helping, since without further information, the reader has to assume that these regions already have low weight due to wpol. As the differentiation between the pollution and the trop. res. weights is not immediately clear to the reader anyways, it might be a good idea to find an example of unusually high polluted regions, which would not have been assigned low weights by using wpol alone.

Reply: The listed regions indeed already have a low $w_{pol}$. However, the additional use of $w_{TR}$ further lowers the net weight by orders of magnitude. While in some regions with $w_{pol} < 1$, $w_{cld}$ lifts the net weight >1 (e.g. over Northern Scotland), $w_{TR}$ keeps the net weight low for

actually polluted pixels. In particular over Eastern China, the measurements with high TVCDs are essentially ignored during weighted convolution (even in case of clouds).

Examples for low $w_{TR}$ outside the potentially polluted regions ($w_{pol}=0$) cannot be found in Fig. 2(c), as they are excluded intentionally. This was a late modification of the algorithm which was accidentally not appropriately updated in the manuscript. So we would like to thank the reviewer for digging deeper in this respect.

The motivation for applying $w_{TR}$ only for pixels in potentially polluted regions was the finding that stratospheric structures regularly result in high TR in remote regions, see e.g. the stripe of enhanced TR in the Indian ocean on $1^{st}$ of July in Fig. 5. If $w_{TR}$ would be applied to this stripe, the respective measurements would be weighted even further down, though they actually represent the stratosphere, and the systematic bias would even be amplified.

In the revised manuscript, we have clarified the procedure:

Page 8 line 6:

"*A high value of T* ∗  likely indicates tropospheric pollution, in particular over potentially polluted regions.*"

Page 9 line 3:

"*4. A $w_{TR}<1$, which is meant to decrease the weight of polluted pixels, is only applied over potentially polluted regions with $w_{pol}<1$. This restriction was introduced to avoid the amplification of stratospheric patterns which are still present in the TR, like the stripe of enhanced TR in the Indian ocean on $1^{st}$ of July in Fig. 5.*"

Page 9 line 5:

"*Observations over these regions are already associated with a low pollution weight. However, due to the additional application of $w_{TR}$, the net weight is lowered further by orders of magnitude, and the respective satellite pixels will hardly contribute to the stratospheric estimate in the next iteration, even in case of high $w_{cld}$.*"

• p. 10, l. 1: If the authors set Wij = 0 in case of measurement gaps, then Vij as defined by the authors is not defined. Shouldn't it be enough to set Cij= 0? Otherwise, please amend Eq. 11 so that it yields a well-defined Vijeverywhere.

Reply: In case of measurement gaps, Vij (the mean VCD of all available obervations) is just not defined. In the revised manuscript, we have added a respective note directly after equation 11.

For the procedure of weighted convolution (equations 12 and 13), however, it is essential to set both Cij and Wij to zero in empty grid boxes.

• p. 10, l. 4: The authors should clarify if the 2D Gaussian they use as CK is defined in degree-space or in kilometer-space. If it is defined in degree-space, they should justify the resulting inconsistency depending on latitude.

Reply: The first part of the section is meant to describe the general procedure of weighted convolution, with a yet unspecified CK, while the details of the choice of CK are given later. To make this clear, we have replaced *""* by *"(see below)"*.

In the definition of G (eq. 15), it is evident that a lat/lon grid is chosen. We have added *"Note that the difference of the CKs, which are defined on a regular degree grid, is even more drastic in kilometer space."* to equation 15.

• p. 13, l. 31: There's a spurious "see" in the refernce to Jöckel et al.

Reply: Corrected.

• p. 14, l. 6: The authors should specify how the EMAC model determines the tropopause height, i.e., thermal, dynamical, ... criterion?

Reply: We have revised the sentence to

*"Stratospheric VCDs were calculated by vertical integration of the modelled $NO_2$ mixing ratios between the tropopause height (as diagnosed according to the WMO definition based on lapse rate equatorwards of 30° North/South, and as iso-surface of 3.5 PVU potential vorticity poleward of 30° latitude) and the top of the atmosphere."*

• p. 17, l. 5: Currently, the manuscript states that "the final Vstrat [...] as weighted mean of both [CKs]". This is not really precise, it should rather say that the final Vstrat is the weighted mean of Vstrat calculated with both CKs.

Reply: We have modified this sentence to

*"In STREAM, two different CK are applied, yielding two stratospheric estimates, and the final $V_{strat}$ is calculated as weighted mean of both (see section 2.3 and equations 15 and 16)."*

• p. 18, l. 6: The authors write of "the" small-scale structures of strat. NO2 in EMAC. It would be good to elaborate a bit on the "the", i.e., in which regions, in which months, ...
I suspect that these structures are mostly there at low
latitudes, but it would be good if the authors could be more explicit about this.

Reply: To illustrate this, we have added a figure showing the synthetic stratospheric and tropospheric NO2 fields to the supplement:

[Figure]

*Figure C : Synthetic $NO_2$ column densities of stratosphere (EMAC, top), troposphere (TM4, center) and total V\* (bottom) for 1 January (left) and 1 July (right) 2005.*

We have extended the discussion accordingly:

*"This is mainly caused by small-scale structures of stratospheric $NO_2$ in EMAC over the Pacific, in particular at southern latitudes, which are resolved by neither STREAM nor RSM (see Figure C)."*

• p. 18, l. 10-11: The authors should clarify if the "remaining biases" are low or

high biases.

Reply: We have specified this sentence to

*"Remaining systematic biases are about -0.1 CDU over polluted regions, i.e. resulting TRs are slightly underestimated, as expected due to the general approach of using total column measurements as proxy for the stratospheric estimation."*

• p. 18, l. 26: "meaningful" has an extra "l"

Reply: Corrected.

• p. 19, l. 11-12: The authors should comment on whether they expect higher variability of T∗ in DOMINO or STREAM.

Reply: STREAM uses convolution to derive stratospheric fields, with large-scale convolution kernels at low latitudes. Thus, any variability of stratospheric NO2 on smaller spatial scales will result in increased variability of T*, even if the mean is appropriately estimated. DOMINO, on the other hand, can generally resolve gradients on smaller scales. However, daily maps of TR from DOMINO (Fig. S19) reveal patchy residuals over the ocean. This might be related to a misrepresentation of stratospheric gradients in the model, but this is not our expertise and not the focus of the manuscript.

• p. 19, l. 17: underestimation "by", not "of" DOMINO

Reply: Corrected.

• p. 21, l. 9-11: Might the longitudinal dependency in STSEMAC partly be caused by the temporal sampling of the EMAC model? The LT difference between East and West end of the OMI swatch is rather large, and using a fixed EMAC time might introduce longitudinally varying biases

Reply: The effect of different local time at the swath edges is generally small, as shown in detail within the reply to the major comment #2 of reviewer 1.

What is meant here is the large scale longitudinal pattern of stratospheric NO2: over the Pacific, $STS_{EMAC}$ is by construction matching to OMI. But the longitudinal dependency at northern latitudes is quite different for STREAM and EMAC (compare the updated Fig. 3, which now includes EMAC stratosphere as well as proposed by reviewer 1). Consequently, STREAM results in low background TR over Siberia (Fig. 5), while $STS_{EMAC}$ underestimates the stratospheric column over Siberia and results in high biased TR (similar as RSM).

We have revised the manuscript accordingly:

*"The systematic deviations North from 35°N (1.&2.) are caused by the longitudinal dependency of stratospheric NO$_2$ from EMAC which differs from the pattern in total column (see Fig. 3). In detail, stratospheric NO$_2$ over Siberia is quite low in EMAC, resulting in high biased TR (similar as for RSM) and indicating that the mean longitudinal dependency of stratospheric NO$_2$ is not fully reproduced by EMAC."*

• p. 21, l. 28-29: GOME/SCIAMACHY/GOME-2 might have systematic differences in the CTP products, which might in turn influence the STS results. Furthermore, the spatial resolution of the different instruments should lead to different pdfs for the cloud fraction, again possibly influencing the STS results. It would be good if the authors could comment on this issue.

Reply: We have modified the sentence to

*"This might be related to the differences of cloud statistics due to pixel size, in particular a lower number of fully clouded pixels for GOME-2, as well as differences in local time, cloud products, or systematic spectral interferences caused by clouds in either retrieval algorithm."*

• p. 22, l. 24: "MOZART" should be replaced with "MOZART-2"
Reply: Corrected.

• p. 24, l. 20: Also, the smaller pixel size will lead to a higher fraction of purely
clouded pixels. The authors should briefly comment on the implications.
Reply: This is already stated in line 20 of the manuscript, with the implication that more
sampling points will become available over potentially polluted regions (line 21). See also the
reply to the comment above, which already picks up this aspect in section 5.2.

• Sect. 5.5.2: Sentinel-4 is the name of the satellite, not the instrument. The
authors should instead write something like "UVN onboard Sentinel-4" or the like.
Reply: Sentinel-4 is the name of the mission, which will include a UVN instrument to be
launched on a MTG-S (Meteosat Third Generation Sounder) satellite.
We have tried to be more precise on this without bothering the reader with details irrelevant
for this study by changing lines 25-26 to
*"Over Europe, Sentinel 4 (S4, Ingmann et al., 2012) will be the first*  *mission
providing a spectral resolving UV/vis instrument on a geostationary satellite."*
For simplicity, we still refer to "S4 measurements" in the remaining section.

• p. 26, l. 8: Please add "on average" before "negative T∗, because single negative
T∗are expected, as the authors already noticed elsewhere.
Reply: Done.

• p. 26, l. 31-32: The authors should note that applying STREAM to other trace
gases, where the bulk of the profile cannot be expected to lie within the boundary
layer but rather in the same altitude ranges as clouds is at least challenging.
Reply: The intended message here was that the concept of weighted convolution might be
useful for background estimates in other contexts as well. Of course, the reviewer is right in
pointing out that the application of $w_{cld}$ is not appropriate if the background in question is of
tropospheric nature, or any kind of algorithm bias.
We have thus revised the last sentence to
*"Thus, the concept of weighted convolution could be used within the satellite retrievals of e.g.
$SO_2$, BrO, HCHO, or CHOCHO, with appropriately chosen and optimized weighting
factors."*

• p. 27, l. 21-24: The authors should think about ways in which
the LT of measurement being in the morning could hinder STREAM for
GOME/SCIAMACHY/GOME-2.
Reply: As demonstrated, the algorithm generally successfully worked for GOME1/2 and
SCIAMACHY, except for the tropospheric background, which turned out to be related to
different dependency of total columns on cloud conditions as compared to OMI.
We thought and discussed among the co-authors about the possible impact of LT on the
observed differences in the response to cloudy pixels. However, we could not think of any
mechanism how the differences in LT could explain our findings. Still, we have modified the
respective sentence to
*"The emphasis of clouded observations, which provide a direct measurement of the
stratospheric rather than the total column, should supersede an additional correction for the
tropospheric background, which successfully worked for OMI, but less for GOME and
SCIAMACHY. This might be related to differences in pixel size or local overpass time, both*

*potentially affecting cloud statistics, or differences in the cloud algorithms. However, the detailed reasons are not yet fully understood and require further investigations.*"

• Fig. 8: I cannot understand how the 90% percentile T∗over polluted regions can still be lower than 1 CDU. Is this really correct? Furthermore, I have trouble deriving the minimum in the T∗difference over China/Japan in Fig. 9 from the statistics given in this Fig. 8.
After seeing the definition of "polluted" in Fig. S7, this becomes clear; but it points to the misleading label "polluted" in this context.
Reply: We agree that the term "*polluted*" is misleading and changed it to "*potentially polluted*" in all respective diagrams.

• Figs 9, 10, 11, S19, S20, S23, S24, S26, S27b, S30-S31, S33 should have ΔT∗ as colorbar label instead of T∗.
Reply: Figures 9-11 display differences of T* from different algorithms, and thus the reviewer is right. We have corrected the labels accordingly.
The listed figures in the supplement, however, actually display T*.

• Fig. S9: Isn't increasing the cloud weight by a factor of 10 equivalent to increasing the total weight by a factor of 10, due to the definition of the total weight in Eq. 8? Why should this Figure then say something about the cloud weight in particular?
Reply: The reviewer is absolutely right: a simple factor of 10 would just raise the total weight everywhere. The description is not exact.
What we have done in the "high cloud" scenario is switching Equation 6(a) from $10^{(2w_c w_p)}$ to $10^{(3w_c w_p)}$. I.e., in case of full cloud cover around 500 hPa ($w_c w_p$=1), $w_{cld}$ is indeed raised by a factor of 10, but for low cloud fraction or for high/low clouds ($w_c w_p$=0), $w_{cld}$ is still 1. We have clarified this in the revised manuscript in section 4.2.1 (b).

• The authors should introduce the OMI instrument before making reference to it from p. 8 onwards. In the current manuscript, OMI is only introduced later, in Sect. 3.1.
Reply: In the revised manuscript, we now introduce SCIAMACHY, GOME-2, and OMI in the first sentence of the introduction:
*"Beginning with the launch of the Global Ozone Monitoring Experiment (GOME) on the ERS-2 satellite in 1995 (Burrows et al., 1999), several instruments (SCIAMACHY, OMI, GOME-2; see Table 1 for acronyms and references) perform spectrally resolved measurements of sunlight reflected by the Earth's surface and atmosphere."*

---

## Author Comment (AC4) · 20 May 2016

**The STRatospheric Estimation Algorithm from Mainz (STREAM): Estimating stratospheric $NO_2$ from nadir-viewing satellites by weighted convolution**

S. Beirle1, C. Hörmann1, P. Jöckel2, S. Liu3, M. Penning de Vries1, A. Pozzer1, H. Sihler1, P. Valks3, and T. Wagner1

[revised manuscript text omitted]

The manuscript is organized in the following way:

In section 2, the STREAM algorithm is described in detail. Section 3 provides information on the satellite and model

25 datasets used in this study. Section 4 analyses the performance of STREAM and its sensitivity to input parameters based on both, actual satellite measurements and synthetic data. In section 5, the STREAM results are discussed in comparison to other STS algorithms, including the TROPOMI prototype algorithm. A general discussion on the challenges and uncertainties of STREAM in particular, and STS in general, is given, followed by conclusions (section 6). Several additional images and tables are provided in a supplementary document.

**30 2 Methods**

STREAM stands in tradition of MRSM algorithms which estimate the stratospheric field directly from satellite measurements for which the tropospheric contribution is considered to be negligible. For this purpose, measurements over remote regions

with negligible tropospheric sources, as well as cloudy measurements are used. In contrast to other MRSMs, however, no strict pollution mask is applied. Instead, weighting factors are used.

STREAM consists basically of two steps:

1. a A set of weighting factors are is calculated for each satellite pixel, determining : a "pollution weight" that reduces the

5 contribution of potentially polluted pixels, a "cloud weight" that increases the contribution of cloudy observations, and a "tropospheric residue weight" that adjusts the total weight in case of exceptionally large or negative tropospheric residues. The product of these weighting factors determines to what extent the measured associated NO2 total columns contribute to the estimated stratospheric field (Sect. 2.2)<del>, and</del>.

[revised manuscript text omitted]

estimate.

10

- A high value of  $T^*$  generally likely indicates tropospheric pollution, in particular over potentially polluted regions. The respective satellite pixels should not be used for the stratospheric estimation.
- As negative columns are nonphysical,  $T^* < 0$  indicates that the stratospheric field has been overestimated. This happens if the weighted convolution with neighbouring pixels with high total columns causes the estimated stratosphere to be even higher than the local total columns. In order to avoid this effect, Consequently, the respective satellite measurements local total columns should be assigned a higher weighting factor such that they contribute more strongly to the stratospheric

We thus define a further weighting factor  $w_{\rm TR}$  which weights down/up the pixels associated with a large positive/negative 20 TR, respectively. It turned out, however, that the stratospheric estimate is very sensitive to the definition of  $w_{\rm TR}$ , and a simple 20 definition based on the TR of individual satellite pixels can easily result in systematic artefacts. This results from  $T^*$  being 27 defined as the difference of  $V^*$  and  $V_{\rm strat}$  (Eq. 3), i.e. two quantities of the same order of magnitude with non-negligible errors. 28 Thus, the resulting statistical distribution of  $T^*$  inevitably includes negative values. These negative values caused by statistical 29 fluctuations must not be excluded from the probability density function in order to keep the mean unbiased, but should not be 29 used as trigger for weighting up the respective measurement within the stratospheric estimation. Thus,  $w_{\rm TR}$  should be only

- applied to significant and systematic deviations of  $T^*$  from 0. This is achieved by the following settings: 1. In contrast to  $w_{cld}$ , which is defined for each individual satellite measurement,  $w_{TR}$  is defined based on the TRs averaged
  - over  $1^{\circ} \times 1^{\circ}$  grid pixels. I.e., first the values of  $T^{*}$  within one grid pixel are averaged, reducing statistical noise, before  $w_{\text{TR}}$  is calculated, and the resulting weight is then assigned to all satellite measurements within the grid pixel.

2.  $w_{\text{TR}}$  is only applied if the absolute value of the mean grid box  $T^*$  exceeds a threshold of 0.5 CDU, which is typically larger than the spectral fitting error:

$$w_{\rm TR} := \begin{cases} 10^{-2 \times T^*} & \text{if } |T^*| > 0.5 \text{ CDU} \\ 1, & \text{else} \end{cases}$$
(7)

5

10

1

wTR is only applied , if a larger area is affected by systematic low or high TR, i.e. if for grid pixels where the adjacent grid pixels exceed the threshold as well. I.e., By this additional condition it is guaranteed that a single outlier will not in the satellite measurements cannot trigger wTR, as any satellite measurement is assigned to one grid pixel (see section 2.3).

4. A  $w_{\text{TR}} < 1$ , which is meant to decrease the weight of polluted pixels, is only applied over potentially polluted regions with  $w_{\text{pol}} < 1$ . This restriction was introduced to avoid the amplification of stratospheric patterns which are still present in the TR, like the stripe of enhanced TR in the Indian ocean on 1st of July in Fig. 5.

[revised manuscript text omitted]

$$G^{\text{pol}} := G(\sigma_{\varphi} = 10^{\circ}, \sigma_{\vartheta} = 5^{\circ})$$

$$G^{\text{eq}} := G(\sigma_{\varphi} = 50^{\circ}, \sigma_{\vartheta} = 10^{\circ})$$
(15)

10

(see Fig. S4 in the supplement). Note that the difference of the CKs, which are defined on a regular degree grid, is even more drastic in kilometer space.

- 2. Stratospheric VCDs  $V_{\text{strat}}^{\text{eq}}$  and  $V_{\text{strat}}^{\text{pol}}$  are derived for both CK according to Eqs. 12-14.
- 3. The final stratospheric VCD is defined as the weighted mean of both, depending on latitude  $\vartheta$ :

$$V_{\text{strat}} := \cos^2(\vartheta) V_{\text{strat}}^{\text{eq}} + \sin^2(\vartheta) V_{\text{strat}}^{\text{pol}}$$
(16)

15 By this method, spatial smoothing is wide enough at the equator (needed to interpolate e.g. the stratosphere over Central Africa), but small enough at the polar vortex.

In latitudinal direction, this procedure can cause small, but systematic biases if stratospheric NO2 shows significant latitudinal gradients on scales of  $\sigma_{lat}$  or smaller. To overcome this, STREAM provides the (default) option to run the weighted convolution on "latitude-corrected" VCDs. I.e., the mean dependency of  $V^*$  on latitude is 1. determined (again over the Pa-

20 cific), 2. subtracted from all individual  $V_{ijk}$ , and 3. added again to the stratospheric estimate from weighted convolution. By this procedure, latitudinal gradients are largely removed for the convolution (but not from the final stratospheric fields), and the systematic biases vanish (as shown in section S2.3).

**2.4 Data processing**

STREAM estimates stratospheric fields and tropospheric residues for individual orbits-, using NO2 measurements of the

- 25 dayside of the orbit. Note that the effect of changes of local time on stratospheric NO2 across orbit is generally low (see section S2.4 in the Supplement) and is thus neglected within STREAM. For each orbit under investigation, the orbit itself plus the 7 previous and subsequent orbits (corresponding to about  $\pm$  12 hours in time, or  $\pm$  180° in space (longitude), for the investigated satellite instruments in polar sun-synchronous orbits) are used for the calculation of  $V^*$ , weighting factors, and thus  $V_{\text{strat}}$  via weighted convolution. For the daily means presented in this study, all orbits where the orbit start date matches
- 30 the day of interest are averaged.

Alternatively, STREAM can be run in "Near-real time" (NRT) mode, in which the 14 past, but no future orbit, are included in the weighted convolution. We discuss the performance of STREAM NRT for the example of GOME-2 in section 5.2.

STREAM v0.92, implemented as MATLAB script at MPI-C, requires about 10 seconds for processing one orbit of OMI data on a normal desktop PC (3.4 GHz). Time consuming steps are, at about equal parts, the sorting of the satellite pixels on

5 the global grid  $V_{ijk}$  (see sect. 2.3), and the convolution process, while the time needed for the calculation of weighting factors is negligible.

**3** Datasets**

**3.1 Satellite datasets**

Several UV/vis satellite instruments provide column measurements of atmospheric NO2. Table 1 summarizes the characteris-

10 tics of the instruments discussed and provides references to the data products used in this study, from which the total  $NO_2$  SCD, the stratospheric AMF, and the cloud fraction/cloud top height are taken as input for STREAM. Below we provide details on the satellite characteristics and the datasets used in this study, starting with OMI (as STREAM was optimized for OMI within TROPOMI verification) and GOME-2, followed by older instrument with particular challenges such as poor spatial coverage (SCIAMACHY) or resolution (GOME).

**15 3.1.1 OMI**

In this study we mainly focus on OMI for two reasons:

- 1. OMI provides daily coverage with small ground pixels. While this already results in a high number of available satellite pixels per day (>106), also the number of clouded pixels matching the requirements to cause a high  $w_{cld}$  is high (more than 105 pixels have a  $w_{cld} > 5$ ).
- 20 2. STREAM is the STS verification algorithm for TROPOMI. Algorithm testing within TROPOMI verification and comparisons to the TROPOMI prototype algorithm are performed based on actual OMI measurements.

STREAM basically requires  $V^*$  (=*S*/*A*strat) as input. For OMI, we use the level 2 "OMNO2" data product (version 3) provided by NASA (Bucsela et al., 2013) and labelled as "Standard Product 2" (SP2) therein, which provides de-striped NO2 SCDs and stratospheric AMFs2. In addition, quality proxies are used to exclude dubious measurements (like those affected by the "row

anomaly"3). Also information on cloud radiance fraction (CRF) and cloud pressure (CP), which is needed for the calculation of  $w_{\text{cld}}$ , is provided by the OMNO2  $\frac{\sqrt{3}}{\sqrt{2003}}$  hdf files, based on the "improved OMI O2-O2 cloud algorithm" (Bucsela et al., 2013) OMCLDO2.

The NASA v3 v003 product involves a STS algorithm based on a MRSM as well. The resulting tropospheric residues of STREAM and NASA v3 v003 are compared and discussed in detail in section 5.1.2.

<sup>2In the DOMINO v2 product, total SCDs are not de-striped, and stratospheric AMFs are only provided up to a SZA of 80°

<sup>3http://projects.knmi.nl/omi/research/product/rowanomaly-background.php#overview

[revised manuscript text omitted]
 The "high  $w_{\text{cld}}$ -" scenario is achieved by modifying Eq. 6(a) from  $w_{\text{cld}} := 10^{2 \times w_c \times w_p}$  to  $w_{\text{cld}} := 10^{3 \times w_c \times w_p}$ . I.e.,

20  $w_{cld}$  is increased by a factor of 10 - for cloudy pixels of mid altitude, but stays unchanged for cloud free pixels. In this scenario, measurements over clouds by far dominate the stratospheric estimate, yielding lower  $V_{strat}$ , and thus higher  $T^*$ , as compared to the baseline. However, the difference is very small (<0.05 CDU). In addition, the variability of  $T^*$  is generally slightly higher in case of a 10 fold increased  $w_{cld}$ .

(c) If high altitude clouds are included in the calculation of  $w_{cld}$ , the resulting TR hardly changes at all, indicating that the 25 impact of lightning NOx on NO2 satellite observations is generally small.

(d) The inclusion of low altitude clouds has almost no effect as well, as expected over clean regions. But over potentially polluted regions, it is expected that low altitude clouds result in increased total columns  $V^*$ , as soon as there is significant NO2 above or within the cloud, causing high tropospheric AMFs. Consequently,  $V_{\text{strat}}$  is expected to be biased high, and  $T^*$  biased low over potentially polluted regions, if low clouds are included in the calculation of  $w_{\text{cld}}$ . This effect was indeed

30 found, but the absolute change is rather small (< 0.1 CDU in winter, almost zero in summer). This weak dependency on the inclusion of low-altitude clouds probably results from the conservative definition of  $w_{pol}$ , being already very low over regularly polluted regions.

Following the argument that cloudy observations provide a direct measurement of the stratospheric column, a higher cloud weight would be expected to be more favourable, and to result in higher tropospheric background over the Pacific. This is indeed observed for OMI. For other satellite instruments, however, results are somewhat contradictory (see sections 5.3, 5.4, and 6). Thus, the definition of  $w_{cld}$  in Eq. 6 is kept as compromise in order to have common algorithm settings across different satellite platforms.

. . .

5

10

15

**4.2.2 Impact of convolution**

In STREAM, two different CK are applied, yielding two stratospheric estimates, and the final  $V_{\text{strat}}$  is calculated as weighted mean of both (see section 2.3 and equations 15&16). We tested the impact of the choice for CK by applying both the polar ("narrow") and equatorial ("wide") CK globally. The narrow CK, and thus the potential range of influence of satellite pixels with high weights, is limited to about 20° in longitude. This potentially leads to biases over continents caused by spatial

interpolation. Thus, the resulting  $T^*$  is (too) low over central Africa. Overall, median  $T^*$  over potentially polluted regions is lower compared to the baseline settings by about 0.1 CDU.

For wide CK, on the other hand, the longitudinal gradients at high latitudes are not resolved any more. Consequently, the spatial variability of daily  $T^*$  at high latitudes is increased by a factor of 2. We conclude that our choice of the combined CK for high and low latitudes is a good compromise for realising weighted convolution.

**4.2.3 Impact of latitude correction**

If the initial correction of the latitudinal dependency of  $V^*$  over the Pacific is omitted, the resulting TR reveals global stripes with negative values around the equator and maxima ( $\approx 0.5$  CDU) at about 30°N/S, both in winter and summer.

**4.2.4 Impact of the number of considered orbits**

20 In STREAM baseline settings, for each orbit, stratospheric estimation is based on the previous and subsequent 7 orbits, corresponding to full global coverage for OMI. Switching this parameter to either  $\pm 14$  or  $\pm 3$  orbits has almost no impact on the resulting TR.

In case of NRT application of STREAM, no subsequent orbits are available, and the previous 14 orbits have to be considered. Also this set-up results in essentially the same  $T^*$  statistics (compare sect. 5.2).

**25 4.2.5 Impact of tropospheric residue weight**

In STREAM v0.92, one iteration for  $w_{TR}$  is applied. If  $w_{TR}$  is omitted, the spread of  $T^*$  slightly increases for high latitudes. A second iteration does not yield a further improvement. Lowering the threshold in Eq. 7 from 0.5 to 0.3 CDU results in slightly lower spread of  $T^*$  at high latitudes in summer.

**4.2.6 Impact of pollution weight**

The impact of pollution weight is investigated by multiplying  $w_{pol}$  (where different from 1, compare Fig. 2(a)) by 0.1 ("low  $w_{pol}$ ") or 10 ("high  $w_{pol}$ "). In the first case, the resulting pollution weight over most continents is below 0.01, while in the second case, it is increased to 1 (meaning that  $w_{pol}$  is switched off) except for industrialized pollution hotspots.

In remote regions, the change of pollution weight has almost no impact. In potentially polluted regions, the impact is only

5 moderate as well. Low  $w_{pol}$  does not differ much from the baseline, as the latter already assigns rather low weighting factors to potentially polluted pixels; a further decrease by factor 0.1 thus does not change much.

Only for high  $w_{pol}$ , a significant change of TR can be seen; in this case, the inclusion of more partly polluted observations causes a high bias in the stratospheric estimate and the resulting TR are biased low by almost 0.1 CDU in winter.

**4.3 Performance for synthetic data**

10 In order to estimate the uncertainties of the STREAM stratospheric estimate (and thus tropospheric residues), we apply the algorithm to synthetic input data, as defined in section 3.3, for which the "true" stratospheric fields and TR are known. Again, a simple RSM is applied as well for comparison.

Figure 7 displays the statistics of the error of  $T^*$ , i.e. the difference  $\Delta$  of estimated and a-priori TR, which equals the difference between the true and the estimated stratospheric VCD, for different regions. The spatial patterns of  $\Delta$  are shown in

15 the Supplement (Fig.  $\frac{$18$220}{$220}$ ).

Over the Pacific, RSM results in TR biased low by 0.1 CDU. With STREAM, the bias is reduced (0.05 CDU), but not completely removed. On 1 Jan 2005,  $\Delta$  shows a variability of almost 0.4 CDU (10th to 90th percentile) for both algorithms. This is mainly caused by the small-scale structures of stratospheric NO2 in EMAC , over the Pacific, in particular at southern latitudes (see Fig. S7), which are resolved by neither STREAM nor RSM. The respective spatial variability of the monthly

20 mean, however, is much lower (about 0.1 CDU).

Again, the simple RSM results in large biases and high variability of  $\Delta$  at high latitudes, which are largely reduced by STREAM.

Overall, the agreement of a-priori and estimated  $T^*$  from STREAM is very good, in particular for monthly means. Remaining systematic biases are about 0.1-0.1 CDU over polluted regions, potentially polluted regions, i.e. resulting TRs are slightly

25 underestimated, as expected due to the general approach of using total column measurements as proxy for the stratospheric estimation.

The application of STREAM to synthetic data thus provides a valuable estimate of the algorithm's accuracy. One might think of using the synthetic data for optimizing the definition of weighting factors being the next step forward. However, we refrain from doing so due to some contradictory results for different instruments. Concretely, the remaining bias in TR for synthetic

30 data of about 0.1 CDU could be further reduced by increasing  $w_{cld}$ . This, however, has adverse effects on SCIAMACHY and GOME results (see sections 5.3 and 5.4).

**5 Comparison to other algorithms and discussion**

[revised manuscript text omitted]

**5.1.2 Comparison to NASA**

The official OMI NO2 product provided by NASA uses a MRSM for STS as well, as decribed in Bucsela et al. (2013). Daily 15 and monthly maps of TR from NASA (OMNO2  $\sqrt{3}$ v003/SP2) are shown in Fig. S22. S20.

The NASA STS corrects for the tropospheric background based on a "fixed model estimate" (Bucsela et al., 2013). Consequently, TR are about 0.1 to 0.3 CDU over clean regions throughout the world.

TR from NASA are impressively smooth even on daily basis. This results from the STS algorithm which, over clean regions, interpretes the full total column interprets the difference between the total column and the (small) modelled tropospheric

20 column as stratospheric column, whenever the quotient of the modelled tropospheric slant column and stratospheric AMF (matching our definition of  $T^*$ ) goes below a threshold of 0.3 CDU. Thus, at Southern high latitudes in July (completely classified as unpolluted in Bucsela), the TR is almost  $0\pm 0$ , i.e. shows no variability at all (compare Fig. 8) just by construction, as all the variability present in the total column was assigned to the stratospheric column -(compare Fig. 3).

While this is probably a reasonable procedure over completely clean regions, we would like to point out that:

- 25 1. The smoothness of NASA TR over oceans is not surprising, as it is reached by construction. In particular, the smooth patterns for TR over oceans allow no conclusion on the NASA STS performance over polluted continental regions, where TR are based on interpolated stratospheric fields, just as in STREAM.
  - 2. The NASA procedure of assigning the total column variability in clean regions completely to the stratospheric estimate also removes any cloud dependency of the TR, which affects applications such as profile retrievals by cloud slicing (e.g., Belmonte Rivas et al., 2014).
- 30

5

3. The NASA procedure runs the risk of labelling episodical NO2 transport events over oceans as stratospheric pattern. Bucsela et al. (2013) perform an automatic "hot spot" identification and elimination scheme to avoid this. But, still, on 1st of January, a NO2 transport event can be seen in the total VCD East of Canada (Fig. 3 top left) which is similar to the "meteorological bomb" described in Stohl et al. (2003). This event is clearly visible in  $T^*$  from STREAM (Fig. 5 top left), but only weakly in  $T^*$  from NASA (Fig. \$22, \$20-top left). The reason for this discrepancy is that the local enhancement of NO2 is partly classified as a stratospheric feature in the NASA product, as illustrated in Fig. \$21, \$21, \$22, \$20-top left).

- 5 Figure 10 displays the differences of the monthly mean TR for January and July 2005. Again, overall agreement is very good: In January, both products agree within 0.1 CDU for 69% of the Earth, and within 0.3 CDU anywhere. In July, agreement within 0.1/0.3 CDU is found for 64%/94% of the Earth (for latitudes below 60°), respectively. Again, the band at 30°S sticks out in the difference map as discussed above. Highest deviations of up to 0.5 CDU, however, are observed over the Sahara. Within the NASA STS, the Sahara is masked out completely, as the high albedo and low cloud fractions result in high tropospheric
- 10 AMFs, such that even low tropospheric VCDs could contribute significantly to the total column. In STREAM, however, large parts of the Sahara are treated as unpolluted and are assigned with w=1. A close check of the stratospheric estimates from STREAM and NASA over the Sahara reveals that the large deviation probably results from both, a high biased  $V_{\text{strat}}$  by STREAM, and a low biased  $V_{\text{strat}}$  by NASA (see Fig. S23 right). S22).

**5.1.3 Comparison to $STS_{EMAC}$**

- 15 We have used the stratospheric 3D mixing ratios provided by EMAC in order to perform a simple model-based STS, similar as in Hilboll et al. (2013). First, the latitude-dependent offset between EMAC and OMI VCDs is estimated over the Pacific (if a multiplicative adjustment is performed, results hardly change). Second, the offset-corrected stratospheric NO2 VCDs is used for global STS. No additional correction for the tropospheric background is performed, such that the mean TR over the Pacific is 0 by construction.
- Daily and monthly maps of TR from  $STS_{EMAC}$  are shown in Fig. \$24. \$23. Daily maps reveal patches of TR from -0.3 CDU up to 0.4 CDU resulting from mismatches in actual and modelled stratospheric dynamics. In the monthly mean, these fluctuations cancel out at large part. Overall, variability (10th-90th percentiles) of  $T^*$  in remote regions was found to be about 0.3-0.4, similar as for DOMINO.

Figure 11 displays the differences of the monthly mean TR for January and July 2005. The overall negative values over ocean are a result of the neglect of the tropospheric background in  $STS_{EMAC}$ . Besides this, most striking features are

- positive deviations (i.e., TR from STSEMAC being higher than from STREAM) over North America and Eurasia in January (up to 0.45 CDU, North from 35°N),
- 2. negative deviations over North America and Eurasia in July (down to -0.45 CDU, North from 35°N), and
- 3. positive deviations over the Sahara, Middle East, India and Western China in July (up to 0.38 CDU).
- 30 The systematic deviations North from 35°N (1.&2.) are caused by the longitudinal dependency of  $T^*$  from STSEMAC stratospheric NO2 from EMAC which differs from the pattern in total column (see Fig. S23)and indicate 3). In detail, stratospheric NO2 over Siberia is quite low in EMAC, resulting in high biased TR (similar as for RSM) and indicating that the mean longitudinal

[revised manuscript text omitted]